# Unleashing the Power of Task-Specific Directions in Parameter Efficient Fine-tuning

**Chongjie Si**[1*]      **Zhiyi Shi**[2*]      **Shifan Zhang**[1]      **Xiaokang Yang**[1]
**Hanspeter Pfister**[2]      **Wei Shen**[1†]

[1]MoE Key Lab of Artificial Intelligence, AI Institute, Shanghai Jiao Tong University
[2]Harvard University
chongjiesi@sjtu.edu.cn      wei.shen@sjtu.edu.cn

## Abstract

Large language models demonstrate impressive performance on downstream tasks, yet requiring extensive resource consumption when fully fine-tuning all parameters. To mitigate this, Parameter Efficient Fine-Tuning (PEFT) strategies, such as LoRA, have been developed. In this paper, we delve into the concept of task-specific directions (TSDs)—critical for transitioning large models from pretrained states to task-specific enhancements in PEFT. We propose a framework to clearly define these directions and explore their properties, and practical utilization challenges. We then introduce a novel approach, LoRA-Dash, which aims to maximize the impact of TSDs during the fine-tuning process, thereby enhancing model performance on targeted tasks. Extensive experiments have conclusively demonstrated the effectiveness of LoRA-Dash, and in-depth analyses further reveal the underlying mechanisms of LoRA-Dash.

## 1 Introduction

Large language models (LLMs) Liu et al. (2019); Qin et al. (2023); He et al. (2021b); Touvron et al. (2023a); Devlin et al. (2018); Touvron et al. (2023b); AI@Meta (2024) have exhibited superior performance in a variety of natural language processing (NLP) tasks, including commonsense reasoning Sap et al. (2020) and natural language understanding Wang et al. (2018). Despite the advantages, the practice of fully fine-tuning these models, which often comprise hundreds of millions to hundreds of billions of parameters, requires substantial computational resources and incurs significant memory costs Touvron et al. (2023a); Ma et al. (2024); Raffel et al. (2020). This extensive resource requirement restricts the practical deployment of LLMs across diverse scenarios.

To address this issue, parameter efficient fine-tuning (PEFT) Zhang et al. (2022); Hu et al. (2021); Si et al. (2024a); Houlsby et al. (2019); Zaken et al. (2021); Si et al. (2024b) has been a focal point of recent advancements in adapting LLMs with minimal computational and GPU memory costs. It aims to optimize the number of adjustable parameters to improve task performance without changing the original structure of the model. Among various PEFT methods, LoRA (low-rank adaptation) Hu et al. (2021) stands out. Since the effectiveness of fine-tuning can be ascribed to the reparameterization of a "low-dimensional manifold" Aghajanyan et al. (2020); Li et al. (2018), it suggests that the changes to linear model weights can be characterized by a low-rank structure. Specifically, for each weight matrix $\mathbf{W} \in \mathbb{R}^{n \times m}$, LoRA models its changes $\Delta\mathbf{W} \in \mathbb{R}^{n \times m}$ using two low-rank matrices, $\mathbf{A} \in \mathbb{R}^{n \times r}$ and $\mathbf{B} \in \mathbb{R}^{r \times m}$, with $r \ll \{n, m\}$ to achieve parameter efficiency. The change $\Delta\mathbf{W}$ is added to the pre-trained weights $\mathbf{W}$, and only $\mathbf{A}$ and $\mathbf{B}$ are updated during training. Due to LoRA's flexibility, its applications have become widespread Rombach et al. (2022); Ma et al. (2024).

LoRA underscores that $\Delta\mathbf{W}$ enhances directions in $\mathbf{W}$ that, although not pivotal for pretraining, are indispensable for specific downstream tasks, referring to these as "task-specific directions". Indeed, we suggest that these directions serve as intuitive representations of the low-dimensional manifold, which indicates the significance of task-specific directions for the success of fine-tuning. However, despite that LoRA refers to or even points out the significance of these directions, no existing

---

[*]Equal contributions.
[†]Corresponding author.

studies have concretely defined or systematically pinpointed them (even in those studies on low-dimensional manifold), let alone outlined strategies for their effective utilization. This not only suppresses LoRA's performance potential but also hinders a deeper understanding of the fundamental mechanisms underlying fine-tuning.

The chief goal of this paper is to bridge this gap and further unleash the potential of task-specific directions in fine-tuning, we accomplished two key advancements, which constitute the **contributions of this paper**[1]: First, we propose a framework to **provide a precise definition of task-specific directions**, exploring the properties and practical utilization challenges of these directions. Second, building on the comprehensive analysis, we **introduce a novel method, LoRA-Dash**, which identifies task-specific directions during training and proactively utilizes their influence. Extensive experiments demonstrate the effectiveness of LoRA-Dash, and detailed analytical insights provide us a more comprehensive understanding of LoRA-Dash.

## 2 BACKGROUND

### 2.1 PARAMETER EFFICIENT FINE-TUNING

LFMs' huge complexity and computational demands with billions of parameters create significant challenges for adapting them to specific downstream tasks Xu et al. (2023); Han et al. (2024). Parameter Efficient Fine-Tuning (PEFT) offers an effective solution by reducing the parameters and memory needed for adaptation to a variety of downstream tasks while maintaining performance levels similar to full fine-tuning Si et al. (2024b); Ding et al. (2023). Current PEFT methods can be roughly divided into three distinct categories Si et al. (2024b); Liu et al. (2024): adapter-based Houlsby et al. (2019); Chen et al. (2022); Luo et al. (2023); He et al. (2021a); Mahabadi et al. (2021); Karimi Mahabadi et al. (2021), prompt-based Lester et al. (2021); Razdaibiedina et al. (2023); Wang et al. (2023); Shi & Lipani (2023); Fischer et al. (2024), and low-rank matrix decomposition-based Hu et al. (2021); Liu et al. (2024); Hyeon-Woo et al. (2021); Qiu et al. (2023); Renduchintala et al. (2023); Kopiczko et al. (2023); YEH et al. (2023); Zhang et al. (2022). The first category of methods enhances performance by integrating linear modules with existing layers, either sequentially or concurrently. The second category focuses on refining trainable vectors by adding soft tokens (prompts) to the initial input. The third type, introduced by LoRA Hu et al. (2021), employs low-rank decomposition to model weight changes during fine-tuning and can be combined with pre-trained weights.

### 2.2 LoRA: LOW-RANK ADAPTATION

Based on findings that updates to the weights typically exhibit a low intrinsic rank Aghajanyan et al. (2020); Li et al. (2018), LoRA models the changes $\Delta \mathbf{W} \in \mathbb{R}^{n \times m}$ for each layer's weights $\mathbf{W} \in \mathbb{R}^{n \times m}$ as $\Delta \mathbf{W} = \mathbf{AB}$, where $\mathbf{A} \in \mathbb{R}^{n \times r}$, $\mathbf{B} \in \mathbb{R}^{r \times m}$ with the rank $r \ll \{n, m\}$ to achieve parameter efficiency. For the original output $\mathbf{h} = \mathbf{Wx}$, the modified forward pass is

$$\mathbf{h} = \mathbf{Wx} + \Delta \mathbf{Wx} = (\mathbf{W} + \mathbf{AB})\mathbf{x}. \tag{1}$$

In the training initialization for LoRA, matrix $\mathbf{A}$ is commonly set with Kaiming distribution He et al. (2015), and matrix $\mathbf{B}$ is initialized with zeros, which sets the initial $\Delta \mathbf{W}$ to zero at beginning. During training, LoRA only updates the low-rank matrices $\mathbf{A}$ and $\mathbf{B}$ with $\mathbf{W}$ being frozen. During inference, the low-rank matrices are integrated into the $\mathbf{W}$, resulting in no additional costs.

### 2.3 RETHINKING "TASK-SPECIFIC DIRECTIONS" IN LoRA

LoRA highlights that $\Delta \mathbf{W}$ encompasses directions within $\mathbf{W}$ that are deemed insignificant for initial tasks but are crucial for specific downstream tasks, which LoRA refers to as "task-specific directions" (TSDs, plural). Indeed, we suggest that "TSDs" are intuitive depictions of the low-dimensional manifold Aghajanyan et al. (2020); Li et al. (2018). Given that the most impactful transformations occur within the low-dimensional manifold during fine-tuning, TSDs are also significant in PEFT. However, the description of TSDs in LoRA, as shown in Sec. B.1, presents several contradictions that contribute to confusion. This confusion complicates the identification of TSDs and the subsequent utilization of them.

---

[1]Please refer to Sec. A for more details on the structure and contributions of this paper.

We believe that these issues are mainly due to an unclear definition of task-specific directions. Consequently, to clearly define TSDs and enhance their practical application, we have opted to discard all previous definitions utilized by LoRA. We aim to undertake a comprehensive reconstruction of the entire framework from scratch, ensuring that TSDs are not only well-defined but also effectively utilized in further applications.

# 3 BUILD A FRAMEWORK: UNDERSTANDING TASK-SPECIFIC DIRECTIONS

## 3.1 PRELIMINARIES

Considering a matrix $\mathbf{A} \in \mathbb{R}^{n \times m}$ where $n < m$, it can be decomposed using SVD as $\mathbf{A} = \mathbf{U\Sigma V}^{\mathsf{T}}$. Here, $\mathbf{\Sigma} = \mathrm{diag}(\sigma_1, \ldots, \sigma_n)$ is the diagonal matrix of singular values, with $\mathbf{U}$ and $\mathbf{V}$ being the left and right singular vectors, respectively. This decomposition can be further expressed as

$$\mathbf{A} = \sum_{i=1}^{n} \sigma_i \mathbf{u}_i \mathbf{v}_i^{\mathsf{T}}. \tag{2}$$

Since $\{\mathbf{u}_i\}$ forms the orthogonal bases for $\mathbb{R}^n$ and $\{\mathbf{v}_i\}$ for $\mathbb{R}^m$, the matrix space $\mathbb{R}^{n \times m}$ can be spanned by the bases $\{\mathbf{u}_i \mathbf{v}_j^{\mathsf{T}}\}$, which are also orthogonal. Thus, matrix $\mathbf{A}$ can be seen as a matrix in a subspace of the original $\mathbb{R}^{n \times m}$ space, spanned by a set of linearly independent bases $\{\mathbf{u}_i \mathbf{v}_i^{\mathsf{T}}\}$.

**Definition 1.** *For a matrix $\mathbf{A}$ with its left and right singular vectors represented by matrices $\mathbf{U}$ and $\mathbf{V}$, respectively, the bases of $\mathbf{A}$ are defined as follows:*

- ***Core Bases**: The core bases of the matrix $\mathbf{A}$ are defined as $\{\mathbf{u}_i \mathbf{v}_i^{\mathsf{T}}\}$, where each $\mathbf{u}_i \mathbf{v}_i^{\mathsf{T}}$ is a rank-one matrix formed by the outer product of singular vectors $\mathbf{u}_i$ and $\mathbf{v}_i$.*

- ***Global Bases**: The global bases of the matrix $\mathbf{A}$ are defined as $\{\mathbf{u}_i \mathbf{v}_j^{\mathsf{T}}\}$ for all $i, j$, covering all combinations of the left and right singular vectors.*

**Definition 2.** *The **direction** of a matrix $\mathbf{A} \in \mathbb{R}^{n \times m}$ (where $n < m$) is defined based on its global bases, using an expanded set of its singular values padded with zeros, specifically as $(\sigma_1, 0, \ldots, 0, \sigma_2, 0, \ldots, 0, \ldots, \sigma_n, \ldots, 0) \in \mathbb{R}^{nm}$, i.e., flattened $\mathbf{\Sigma}$ by rows.*

Note that any global basis can be regarded as a direction of unit, since its direction is a one-hot vector. For simplicity in further discussions, we will not differentiate between global bases (core bases) and global directions (core directions).

## 3.2 WHAT ARE TASK-SPECIFIC DIRECTIONS?

We start with the fact that for any specific task, there exists an optimal matrix $\mathbf{W}^* \in \mathbb{R}^{n \times m}$ within the matrix space $\mathbb{R}^{n \times m}$ Si et al. (2024b). For a pretrained matrix $\mathbf{W}$, its optimal alteration for this specific task is defined as $\Delta \mathbf{W}^* = \mathbf{W}^* - \mathbf{W}$. In PEFT, we only possess the information of $\mathbf{W}$ and the directions of $\mathbf{W}$. Since $\Delta \mathbf{W}^*$ and $\mathbf{W}^*$ are established on their respective bases, we initially project both $\Delta \mathbf{W}^*$ and $\mathbf{W}^*$ onto the global bases of $\mathbf{W}$, capturing their directions.

**Definition 3.** *Define $\mathbf{\Pi}.(\cdot)$ as a projection operator that projects a direction in one coordinate system onto another coordinate system. Specifically, $\mathbf{\Pi_W}(\mathbf{A}) = (p_{11}, \ldots, p_{nm}) \in \mathbb{R}^{nm}$ is the projection of the direction of a matrix $\mathbf{A} \in \mathbb{R}^{n \times m}$ onto the global bases of another matrix $\mathbf{W} \in \mathbb{R}^{n \times m}$. $p_{ij} = \mathbf{u}_i^{\mathsf{T}} \mathbf{A} \mathbf{v}_j$ where $\mathbf{u}_i$ and $\mathbf{v}_j$ are the left and right singular vectors of $\mathbf{W}$, respectively.*

Established on the global bases of $\mathbf{W}$, $\mathbf{\Pi_W}(\mathbf{W}^*)$ represents the direction that $\mathbf{W}$ should evolve. $\mathbf{W}$ itself can only utilize up to $n$ bases (i.e., its core bases), meaning that it can at most alter $n$ values of its direction. Thus, we only focus on the changes related to its core directions (for more details, please refer to Sec. B.3). During the transformation, not all core directions' coordinate values change to the same extent. Given the diversity of downstream tasks and the variability in $\mathbf{W}^*$ for different tasks, certain core directions of $\mathbf{W}$ undergo significant changes while others may experience only minimal alterations. Specifically, the change rate $\delta_i$ of the coordinate value corresponding to the $i$-th core direction $\mathbf{u}_i \mathbf{v}_i^{\mathsf{T}}$ of $\mathbf{W}$ is represented as:

$$\delta_i = \left| \frac{\mathbf{\Pi_W}(\mathbf{W}^*)_{ii} - \sigma_i}{\sigma_i + \epsilon} \right| = \left| \frac{\mathbf{u}_i^{\mathsf{T}} \mathbf{W}^* \mathbf{v}_i - \mathbf{u}_i^{\mathsf{T}} \mathbf{W} \mathbf{v}_i}{\sigma_i + \epsilon} \right| = \left| \frac{\mathbf{u}_i^{\mathsf{T}} \Delta \mathbf{W}^* \mathbf{v}_i}{\sigma_i + \epsilon} \right| = \left| \frac{\mathbf{\Pi_W}(\Delta \mathbf{W}^*)_{ii}}{\sigma_i + \epsilon} \right|. \tag{3}$$

Here, $\epsilon = 10^{-6}$ is a constant to prevent singular values from being zero. $\sigma_i$ is the $i$-th singular value of the matrix $\mathbf{W}$, and $\delta_i$ represents the change rate of the coordinate value required in the $i$-th core direction $\mathbf{u}_i\mathbf{v}_i^\mathsf{T}$ for transformation from $\mathbf{W}$ to $\mathbf{\Pi}_{\mathbf{W}}(\mathbf{W}^*)$, specifically quantifies the extent of adaptation needed for a particular task. Hence, we finally define the TSD as:

**Definition 4.** *For a specific task and a pre-trained weight* $\mathbf{W}$*, considering the optimal weights for this task as* $\mathbf{W}^*$*, the **task-specific directions (TSDs)** of this task on* $\mathbf{W}$ *are* $\mathbf{W}$*'s core directions whose coordinate values exhibit significantly higher change rates* $\delta$ *through the alteration from* $\mathbf{W}$ *to* $\mathbf{W}^*$*.*

Inspired by Eq. (3) and Definition 4, it is obvious that $\delta$ is directly related to the projection of $\Delta\mathbf{W}^*$. We are now also equipped to revisit and more precisely articulate LoRA's conclusions on TSDs:

- TSDs are the subset of core directions of pretrained weights $\mathbf{W}$. They are specific to each task, meaning that they vary from one task to another but remain fixed for a given task.

- Core directions associated with larger singular values are less likely to be identified as TSDs, as the change rates of their coordinate values are typically smaller than those associated with smaller singular values. It is reasonable since that larger singular values usually encapsulate more generalized information that the model acquired during pretraining.

We encourage readers to refer to Secs. B.2-B.3 for more detailed reasons on Definition 4.

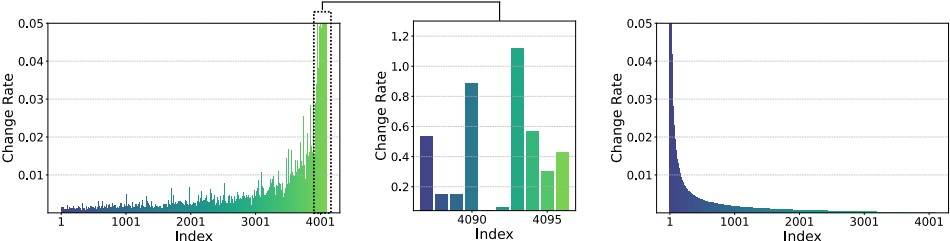

Figure 1: **Left**: The change rates of $\mathbf{W}$'s core bases, based on $\mathbf{W}^*$, vary significantly across each basis, and the bases with higher change rates tend to be concentrated towards the end. **Middle**: The change rates for the directions corresponding to the smallest 10 singular values of $\mathbf{W}$. The directions associated with the smallest singular values do not always exhibit the highest change rates. **Right**: After sorting the change rates from highest to lowest, it is evident that only a few directions have significant change rates, while most exhibit very low change rates. The weights are taken from the 16th layer of LLaMA-7B, and the change rates are scaled (Sec. D.2 for more details).

### 3.3 WHAT ARE THE PROPERTIES OF TASK-SPECIFIC DIRECTIONS?

To further explore the properties of TSDs, we fully fine-tune LLaMA-7B Touvron et al. (2023a) on commonsense reasoning tasks, and assume that $\mathbf{W}^*$ can be obtained through fully fine-tuning. After acquiring the fully fine-tuned weights $\mathbf{W}^*$, we compute $\Delta\mathbf{W}^* = \mathbf{W}^* - \mathbf{W}$, and based on $\Delta\mathbf{W}^*$, we calculate the change rates of core directions of $\mathbf{W}$. The results are shown in Fig. 1, where we can draw several conclusions (Please refer to Sec. D.2 for more details):

- TSD predominantly corresponds to core directions associated with smaller singular values of $\mathbf{W}$, though not the smallest.

- TSD encompasses only few directions where substantial change rates occur; most other core directions exhibit minimal or negligible change rates.

### 3.4 CHALLENGES AND EXPLORATIONS: USING TSD IS NOT AS EASY AS EXPECTED

In downstream tasks, TSDs represent the directions of greatest change, making them critical targets for fine-tuning to adapt pre-trained models effectively to new tasks. Although we have thoroughly explored the definition and properties of TSDs, a significant challenge is that both $\Delta\mathbf{W}^*$ and $\mathbf{W}^*$

are unknown before fine-tuning, indicating that utilizing TSD information beforehand in practical fine-tuning scenarios is nearly impossible.

Despite the apparent challenges, we place our confidence in the potential of $\Delta\mathbf{W}$. We hypothesized that the core directions with the highest change rates predicted by $\Delta\mathbf{W}$ from LoRA are strongly associated with TSDs. To test this hypothesis, we conducted extensive experiments. We strongly encourage readers to refer to Sec. D.3 for detailed experimental setups. The results are shown in the Figs. 2 and 8, where we observe a significant overlap between the predicted directions and the actual TSD, highlighting their strong correlation. It leads to an important conclusion:

**Observation 1.** *Irrespective of the rank setting in LoRA, the training step, or the specific layer within the model, LoRA's $\Delta\mathbf{W}$ consistently captures the information of the task-specific directions.*

This is an exhilarating conclusion, suggesting that even without prior knowledge of TSDs, we can still capture their crucial information through the $\Delta\mathbf{W}$ obtained during LoRA training!

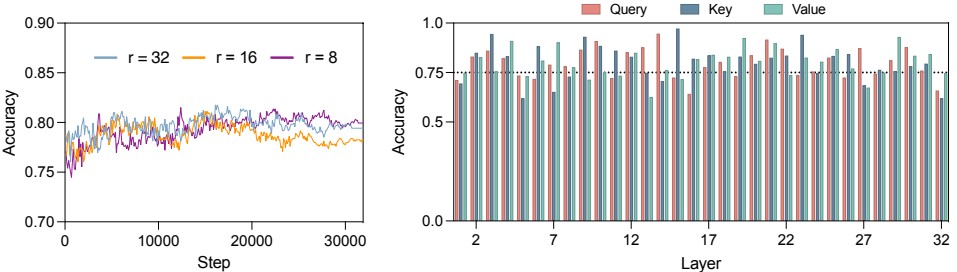

Figure 2: We track the accuracy of predicted directions every 100 training steps in the query, key and value layers of the LLaMA-7B during LoRA fine-tuning, analyzing how well the continuous updated $\Delta\mathbf{W}$ captures TSDs. **Left**: We compute the average accuracy across all query, key, and value layers, showing the model's ability to retain task-specific knowledge for each training step. Across various rank settings of LoRA, these accuracy rates consistently exceed 0.75, indicating that $\Delta\mathbf{W}$ reliably captures and integrates TSD information. **Right**: For a rank setting of $r = 32$, we compute the average accuracy across all steps for each query, key and value layers, revealing their sensitivity to TSDs. The majority of layers maintain an average accuracy above 0.75, showing the robustness to capture TSD information.

# 4 LoRA-Dash: Unleash the Power of Task-specific Directions

To further unleash the power of TSDs in downstream tasks, this section introduces a novel method: LoRA-Dash. LoRA-Dash consists of two principle phases: the "pre-launch phase", where TSDs are identified, and the "dash phase", which unleash the potential of theses identified TSDs.

## 4.1 Pre-Launch Phase

In this phase, LoRA-Dash initially trains the matrices $\mathbf{A}$ and $\mathbf{B}$ (i.e., LoRA's $\Delta\mathbf{W}$) to capture the information of TSDs. To expedite the utilization of TSD information and based on Proposition 1, we posit that there is a predefined number of training steps, $t$, after which $\Delta\mathbf{W}$ can consistently capture TSD information at any stage of training. We set $t$ to 100 as suggested in Sec. 3.4.

Subsequently, we project $\Delta\mathbf{W}$ onto the core directions of $\mathbf{W}$ to calculate the change rates:

$$\delta_i = \frac{\mathbf{u}_i^\top \Delta\mathbf{W} \mathbf{v}_i}{\sigma_i + \epsilon}, \tag{4}$$

where $\mathbf{u}_i$, $\mathbf{v}_i$, and $\sigma_i$ represent the $i$-th left and right singular vectors and singular value of $\mathbf{W}$, respectively. We identify the top $s$ core directions with the highest change rates, setting $s$ as another hyper-parameter, which is recommended to be 8 as suggested in Sec. 3.4. We denote $\bar{\mathbf{u}}_i\bar{\mathbf{v}}_i^\top$ for

$i = 1, 2, \ldots, s$ as the identified "TSDs" for dash, To differentiate from the true TSDs derived from $\mathbf{W}^*$ (or theoretically defined), we label these directions as "launched TSDs" (LTSDs)[2].

## 4.2 DASH PHASE

To leverage the identified LTSDs, we directly learn the changes in their coordinates, denoted as $\Delta\sigma_i$ for the $i$-th $\bar{\mathbf{u}}_i\bar{\mathbf{v}}_i^\mathsf{T}$, to further fine-tune the model for downstream tasks. $\Delta\sigma_i$ is initialized as zero. Mathematically, this adjustment is represented by

$$\sum_{i=1}^{s} \Delta\sigma_i \bar{\mathbf{u}}_i \bar{\mathbf{v}}_i^\mathsf{T}. \tag{5}$$

Ultimately, the updated weight matrix is given by:

$$\mathbf{W} + \Delta\mathbf{W}_{all} = \mathbf{W} + \Delta\mathbf{W}_{AB} + \Delta\mathbf{W}_{dash} = \mathbf{W} + \mathbf{AB} + \sum_{i=1}^{s} \Delta\sigma_i \bar{\mathbf{u}}_i \bar{\mathbf{v}}_i^\mathsf{T}. \tag{6}$$

This equation forms the basis of LoRA-Dash, which aims to fully utilize the power of LTSDs derived from LoRA. During training, $\mathbf{A}$, $\mathbf{B}$, and $\Delta\sigma_i$ are continuously updated. For more details on LoRA-Dash, please refer to Sec. C.

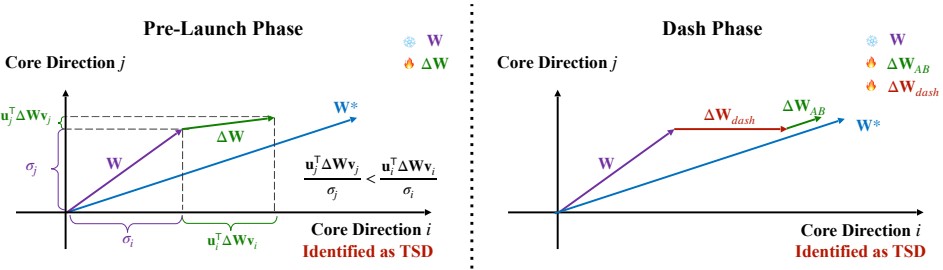

Figure 3: Framework of LoRA-Dash. LoRA-Dash comprises two phases: the pre-launch phase for identifying TSDs, and the dash phase to maximize their potential.

## 5 HOW MUCH CAN TSD BOOST THE PERFORMANCE OF LORA-DASH?

To explore the performance gains of LoRA-Dash, both numerical and visual comparisons are employed. The numerical experiments focus on commonsense reasoning and natural language understanding tasks, assessing LoRA-Dash's performance improvements in standard natural language processing metrics. Visual comparisons focuses on subject-driven generation task, involving using a text-to-image diffusion model to learn specific concepts (i.e., task-specific concept), providing a clear, qualitative and intuitive understanding of the impact of TSDs on performance. The implementation details are shown in Sec. D.1. The experiment details are shown in Secs. D.4-D.6.

## 5.1 NUMERICAL RESULTS

The numerical results, as shown in Tables. 1-2, demonstrate that LoRA-Dash significantly outperforms LoRA across all tasks and models. This distinction is so pronounced that further elaboration on performance gains might be unnecessary. Therefore, we instead focus on several key insights derived beyond mere performance gains.

**Robustness to Parameter Budget:** (When fine-tuning LLaMA-7B) LoRA shows enhanced performance with an increased parameter budget, indicating its dependence on larger rank sizes. Conversely, LoRA-Dash maintains strong performance even under limited parameter conditions, which

---

[2]As shown in Secs. 3.4 and 6.1, LTSDs and TSDs are heavily overlapped. Given that LTSD closely represents TSD, unless explicitly emphasized otherwise, we treat LTSD as equivalent to TSD for analytical purposes.

Table 1: Results on commonsense reasoning tasks. We fine-tune LLaMA-7B Touvron et al. (2023a), LLaMA2-7B Touvron et al. (2023b) and LLaMA3-8B AI@Meta (2024) on this task.

| Model | Method | Params(%) | BoolQ | PIQA | SIQA | HellaS. | WinoG. | ARC-e | ARC-c | OBQA | Avg. |
|---|---|---|---|---|---|---|---|---|---|---|---|
| ChatGPT | - | - | 73.1 | 85.4 | 68.5 | 78.5 | 66.1 | 89.8 | 79.9 | 74.8 | 77.0 |
| LLaMA-7B | Fully FT | 100 | 69.9 | 84.2 | 78.9 | 92.3 | 83.3 | 86.6 | 72.8 | 83.4 | 81.4 |
| | LoRA$_{r=4}$ | 0.10 | 2.3 | 46.1 | 18.3 | 19.7 | 55.2 | 65.4 | 51.9 | 57.0 | 39.5 |
| | LoRA-Dash | 0.10 | 65.2 | 79.9 | 78.3 | 82.8 | 77.1 | 78.6 | 65.4 | 78.4 | **75.7** |
| | LoRA$_{r=8}$ | 0.21 | 31.3 | 57.0 | 44.0 | 11.8 | 43.3 | 45.7 | 39.2 | 53.8 | 40.7 |
| | LoRA-Dash | 0.21 | 69.8 | 81.1 | 77.3 | 85.1 | 81.1 | 77.2 | 64.1 | 79.6 | **76.9** |
| | LoRA$_{r=16}$ | 0.42 | 69.9 | 77.8 | 75.1 | 72.1 | 55.8 | 77.1 | 62.2 | 78.0 | 70.9 |
| | LoRA-Dash | 0.42 | 66.9 | 80.2 | 77.8 | 78.8 | 79.2 | 78.0 | 61.9 | 77.4 | **75.0** |
| | LoRA$_{r=32}$ | 0.83 | 68.9 | 80.7 | 77.4 | 78.1 | 78.8 | 77.8 | 61.3 | 74.8 | 74.7 |
| | LoRA-Dash | 0.83 | 69.9 | 82.8 | 78.6 | 84.9 | 81.6 | 82.3 | 66.5 | 80.8 | **78.4** |
| | LoRA$_{r=64}$ | 1.66 | 66.7 | 79.1 | 75.7 | 17.6 | 78.8 | 73.3 | 59.6 | 75.2 | 65.8 |
| | LoRA-Dash | 1.66 | 69.6 | 79.5 | 76.0 | 82.8 | 75.8 | 81.5 | 64.7 | 81.0 | **76.4** |
| LLaMA2-7B | Fully FT | 100 | 72.2 | 84.9 | 80.9 | 93.1 | 84.7 | 87.5 | 74.2 | 85.1 | 82.8 |
| | LoRA$_{r=16}$ | 0.41 | 71.7 | 81.6 | 79.5 | 89.5 | 81.9 | 82.9 | 67.9 | 79.6 | 79.3 |
| | LoRA-Dash | 0.41 | 70.9 | 82.2 | 80.5 | 90.2 | 80.1 | 83.5 | 68.9 | 80.8 | **79.6** |
| | LoRA$_{r=32}$ | 0.82 | 69.8 | 79.9 | 79.5 | 83.6 | 82.6 | 79.8 | 64.7 | 81.0 | 77.6 |
| | LoRA-Dash | 0.82 | 71.0 | 75.7 | 79.3 | 91.1 | 78.6 | 84.2 | 69.8 | 78.8 | **78.6** |
| LLaMA3-8B | Fully FT | 100 | 75.3 | 89.9 | 81.5 | 95.8 | 87.6 | 91.6 | 79.3 | 87.4 | 86.1 |
| | LoRA$_{r=16}$ | 0.35 | 72.3 | 86.7 | 79.3 | 93.5 | 84.8 | 87.7 | 75.7 | 82.8 | 82.8 |
| | LoRA-Dash | 0.35 | 74.8 | 88.0 | 80.6 | 95.2 | 85.6 | 89.0 | 77.4 | 84.8 | **84.4** |
| | LoRA$_{r=32}$ | 0.70 | 70.8 | 85.2 | 79.9 | 91.7 | 84.3 | 84.2 | 71.2 | 79.0 | 80.8 |
| | LoRA-Dash | 0.70 | 75.3 | 88.5 | 80.2 | 95.7 | 86.8 | 90.7 | 80.2 | 85.6 | **85.4** |

Table 2: Results with DeBERTaV3 fine-tuned on GLUE development set. "FT" represents fully fine-tuning, and "Base' and "Large" represent DeBERTaV3-base and DeBERTaV3-large, respectively.

| Method | Params(%) | MNLI Acc | SST-2 Acc | CoLA Mcc | QQP Acc | QNLI Acc | RTE Acc | MRPC Acc | STS-B Corr | All Avg. |
|---|---|---|---|---|---|---|---|---|---|---|
| Base(FT) | 100% | 89.90 | 95.63 | 69.19 | 91.87 | 94.03 | 83.75 | 90.20 | 91.60 | 88.27 |
| LoRA$_{r=2}$ | 0.18% | 90.03 | 93.92 | 69.15 | 90.61 | 93.37 | 87.01 | 90.19 | 90.75 | 88.13 |
| LoRA-Dash | 0.18% | 90.14 | 95.42 | 72.41 | 91.65 | 94.36 | 89.89 | 91.67 | 91.64 | **89.65** |
| LoRA$_{r=8}$ | 0.72% | 89.80 | 93.69 | 69.30 | 91.78 | 92.97 | 86.28 | 90.68 | 91.62 | 88.27 |
| LoRA-Dash | 0.72% | 90.55 | 96.99 | 70.78 | 92.39 | 94.22 | 88.09 | 91.18 | 91.91 | **89.52** |
| Large(FT) | 100% | 91.81 | 96.93 | 75.27 | 93.01 | 96.02 | 92.68 | 92.20 | 92.98 | 91.36 |
| LoRA$_{r=2}$ | 0.20% | 91.33 | 95.87 | 73.89 | 91.84 | 95.14 | 91.69 | 90.68 | 92.85 | 90.41 |
| LoRA-Dash | 0.20% | 91.65 | 96.11 | 76.11 | 92.61 | 95.52 | 92.78 | 92.18 | 93.05 | **91.25** |
| LoRA$_{r=8}$ | 0.80% | 91.38 | 96.33 | 74.48 | 92.54 | 95.48 | 92.05 | 91.17 | 92.92 | 90.79 |
| LoRA-Dash | 0.80% | 91.18 | 96.45 | 76.88 | 92.81 | 95.85 | 93.51 | 91.91 | 92.86 | **91.43** |

suggests that TSDs are crucial for maximizing fine-tuning efficiency. This highlights its effectiveness in harnessing the full potential of TSDs, even in constrained settings.

**Surpassing Fully Fine-Tuning Performance:** While LoRA typically falls short of the results achieved by fully fine-tuning, LoRA-Dash in some cases outperforms fully fine-tuning. This intriguing observation may provide new insights into optimizing training strategies for fine-tuning.

For comparisons with other methods, please refer to Tables. 23-24.

## 5.2 VISUAL RESULTS

The visual comparisons, as shown in Fig. 4, demonstrate the superior fidelity of images generated by LoRA-Dash in aligning with the subjects of the input images compared to those by standard LoRA. For instance, images generated by LoRA of a dog and a vase showed significant deviations from the input, whereas LoRA-Dash's outputs maintained high consistency with the original images.

Moreover, the effectiveness of TSDs is particularly evident in the response of LoRA-Dash to given prompts. TSDs effectively encapsulate the specific information of each prompt, enabling LoRA-

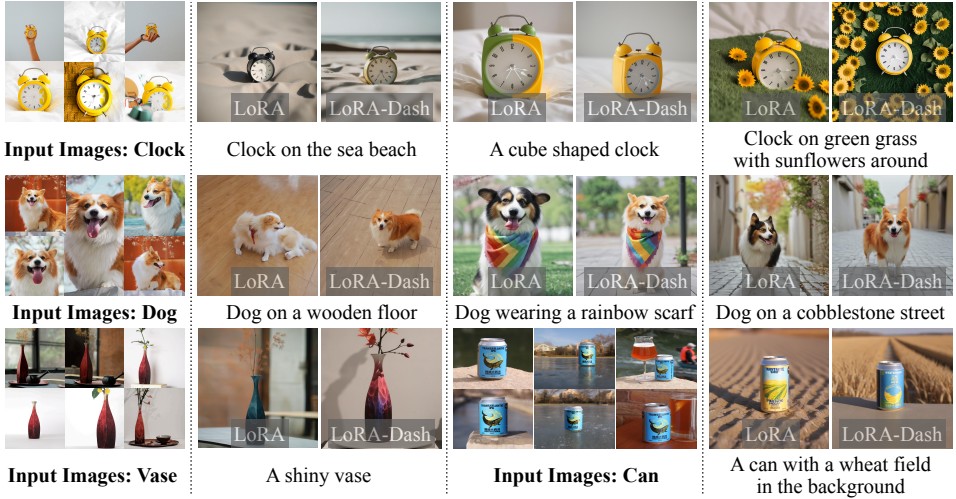

Figure 4: Comparison of generated images from LoRA and LoRA-Dash on subject-driven generation task. LoRA-Dash consistently aligns more closely with the subjects in the input images and adheres better to the given prompts than LoRA.

Dash to precisely grasp the semantic essence of prompts such as *sea beach* or *wheat field*. This capability allows LoRA-Dash to adeptly render complex themes and details within the images.

## 6 UNDERSTANDING LORA-DASH

In this section, we conduct extensive experiments to better understand LoRA-Dash. We first explore the relationships between the LTSDs, the final directions after the dash phase, and TSDs to determine whether LoRA-Dash can amplify the information of TSDs (Sec. 6.1) and, if so, to what extent (Sec. 6.2). We then compare the effectiveness of different directions, such as the top singular directions (Sec. 6.3), and investigate the impact of the length of the pre-launch phase and the number of directions in the dash phase on performance (Sec. 6.4). Finally, we verify whether the information in TSDs can help enhance the effectiveness of other methods (Sec. 6.5). We have also provided extensive discussions in the supplementary material, and we strongly encourage readers to consult it for a more comprehensive understanding of LoRA-Dash.

### 6.1 HOW DOES LORA-DASH DETERMINE AND ALIGN WITH TASK-SPECIFIC DIRECTIONS

We examine the alignment between the LTSDs identified during the pre-launch phase, the directions with highest change rates pinpointed by the final trained $\Delta\mathbf{W}_{all}$ of LoRA-Dash (denoted as "DTSDs", Delta TSDs), and the true TSDs derived from $\mathbf{W}^*$. Following the settings in Secs. 3.4 and D.3, we report the percentage that:

- How many of the top 4 DTSDs directions are contained in $s$ LTSDs[3]. We denote this percentage as "DTSDs ∩ LTSDs " for further convenience.
- How many of the top 4 TSDs are contained in $s$ LTSDs. Denote this as "TSDs ∩ LTSDs"
- How many of the top 4 TSDs are in $s$ DTSDs. Denote this as "TSDs ∩ DTSDs".

The first is to explore the consistency between the directions selected during the pre-launch phase of LoRA-Dash and those showing the greatest change after the final training, and the second and third aim to validate the consistency between the directions chosen during the LoRA-Dash pre-launch phase and the final trained with the true TSDs. We conducted tests on LLaMA-7B Touvron et al. (2023a) and DeBERTaV3-large He et al. (2021b), each with three ranks of LoRA-Dash, and the results are shown in Tables. 3 and 21-22.

---

[3]Here since $s = 8$, the setting is equal to that in Sec. D.3.

Table 3: Alignment on the directions of LoRA-Dash and TSDs. We report the percentage from the self-attention value (v) projection of the first, middle and last layer of two models, as well as the average percentage of all q, k, and v modules. More results are shown in Tables. 21-22.

| Model | LLaMA-7B ($r = 16$) | | | | DeBERTaV3-large ($r = 8$) | | | |
|---|---|---|---|---|---|---|---|---|
| Layer | First (1st) | Middle (16th) | Last (32nd) | All. | First (1st) | Middle (12nd) | Last (24th) | All. |
| DTSDs ∩ LTSDs | 0.75 | 1.00 | 0.75 | 0.77 | 1.00 | 1.00 | 1.00 | 0.98 |
| TSDs ∩ LTSDs | 0.75 | 0.75 | 1.00 | 0.74 | 1.00 | 1.00 | 1.00 | 0.84 |
| TSDs ∩ DTSDs | 1.00 | 0.75 | 1.00 | 0.82 | 1.00 | 1.00 | 1.00 | 0.89 |

It is obvious that LoRA-Dash indeed captures a great proportion of TSDs' information. Moreover, we can also obtain two significant findings: First, the LTSDs and DTSDs of LoRA-Dash align closely, which indicates that *the directions amplified by LoRA-Dash finally are essentially consistent with those selected during the pre-launch phase*. Therefore, since LTSDs contain substantial information about TSDs, it is not surprising that the final DTSDs consistently include information about the true TSDs. Second, when compared with the results in Fig. 8 where the end of each line indicates the final percentage of TSDs ∩ DTSDs of LoRA, it becomes evident that *LoRA-Dash's trained weights capture a greater proportion of TSD information than vanilla LoRA*.

## 6.2 HOW MUCH DOES LORA-DASH AMPLIFY TASK-SPECIFIC DIRECTIONS?

Since LoRA-Dash captures the information of TSDs, we further explore to what extent LoRA-Dash can amplify the information of these directions. To quantify this amplification, we define an amplification factor that measures how significantly LoRA-Dash enhances the task-specific features. The details are shown in Sec. D.7.

Table 4: Amplification on task-specific features of LoRA-Dash. We report the amplification factor from the self-attention key (k) projection of the first, middle and last layer of LLaMA-7B, as well as the average of all q, k, and v modules.

| Layer | $r = 8$ | | | $r = 16$ | | | $r = 32$ | | |
|---|---|---|---|---|---|---|---|---|---|
| | $\Delta\mathbf{W}_{all}$ | $\Delta\mathbf{W}_{AB}$ | $\Delta\mathbf{W}_{dash}$ | $\Delta\mathbf{W}_{all}$ | $\Delta\mathbf{W}_{AB}$ | $\Delta\mathbf{W}_{dash}$ | $\Delta\mathbf{W}_{all}$ | $\Delta\mathbf{W}_{AB}$ | $\Delta\mathbf{W}_{dash}$ |
| 1st | 282.15 | 41.01 | 275.59 | 628.60 | 169.96 | 577.64 | 42.00 | 13.83 | 36.71 |
| 16th | 11.15 | 4.64 | 9.87 | 13.07 | 7.60 | 11.38 | 2.16 | 1.57 | 1.92 |
| 32nd | 68.02 | 7.58 | 66.81 | 30.87 | 11.37 | 29.00 | 8.19 | 1.69 | 8.42 |
| Avg. | 22.85 | 7.33 | 21.07 | 32.12 | 13.36 | 28.44 | 5.28 | 2.53 | 4.57 |

We conduct this experiment on LLaMA-7B with three ranks. We also test the separate impacts of $\Delta\mathbf{W}_{AB}$ and $\Delta\mathbf{W}_{dash}$ on the amplification. The results shown in Table. 4 demonstrate that *LoRA-Dash significantly enhances the features associated with LTSDs under all test conditions*. Notably, the feature amplification contributed by $\Delta\mathbf{W}_{dash}$ accounts for the majority of the overall enhancement, whereas the amplification from $\Delta\mathbf{W}_{AB}$ is relatively minor in comparison. *LoRA-Dash effectively learns and amplifies the majority of TSD information with only $s$ parameters (i.e., $\Delta\sigma_i$ in $\Delta\mathbf{W}_{dash}$), reducing the learning burden on $\Delta\mathbf{W}_{AB}$ during fine-tuning*. This efficiency also allows $\Delta\mathbf{W}_{AB}$ to potentially focus on learning other significant features, further optimizing the model's performance on downstream tasks. We strongly encourage readers to refer to Sec. D.7 for a deeper discussion on this topic.

## 6.3 THE EFFECTIVENESS OF TSD COMPARED WITH OTHER CORE DIRECTIONS

We investigated the effects of TSDs during the pre-launch phase compared to other locations, such as selecting top, bottom or random core directions. The results when fine-tuning LLaMA3-8B, as shown in Fig. 5(a), clearly indicate that selecting TSDs yields the best performance, followed by dash-bottom, with top selections performing the worst. This is not difficult to understand: directly dashing TSDs maximizes the ability of **W** to adapt to downstream tasks. Since TSDs are generally

located towards the end of the spectrum, choosing directions from the bottom naturally results in better outcomes than selecting from the top. We have a more thorough discussion in Sec. E.3.

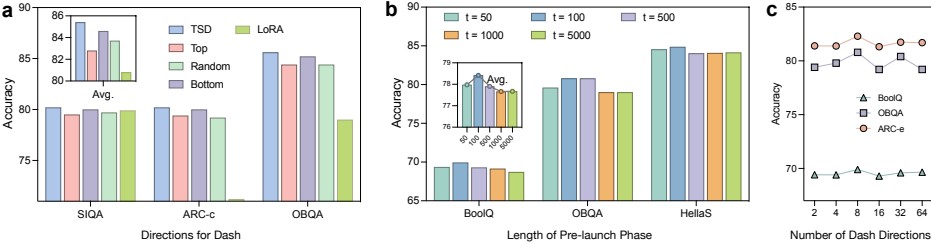

Figure 5: Ablation study of LoRA-Dash. **a**. Ablation study on the directions for dash. **b**. Ablation study on the length of pre-launch phase $t$. **c**. Ablation study on the number of directions $s$ for dash.

## 6.4 THE INFLUENCE OF THE HYPER-PARAMETERS IN LORA-DASH

We evaluate the influence of two hyper-parameters $t$ and $s$ in LoRA-Dash when fine-tuning LLaMA-7B, and the results are illustrated in Fig. 5(b)-(c). It can be observed that starting the dash phase after a longer pre-launch period tends to yield slightly inferior results compared to starting it earlier. This could be attributed to the fact that entering the dash phase earlier allows for more extensive utilization of TSDs, potentially enhancing the adaptation to downstream tasks more effectively. Additionally, a larger value of $s$, representing the number of directions used in the dash phase, also impacts performance. Based on our analysis in Sec. D.7, we suspect that the inclusion of some irrelevant directions might introduce noise into the training process (i.e., destabilizing model convergence), leading to diminished performance.

## 6.5 CAN TSD ENHANCE THE PERFORMANCE OF OTHER METHODS?

To investigate whether TSDs are beneficial for other PEFT methods, we extended our experiments to include AdaLoRA Zhang et al. (2022) and FLoRA Si et al. (2024a), assessing the impact of leveraging TSDs within these frameworks. The results, as shown in Table. 5, clearly indicate that incorporating TSDs significantly enhances the performance of these methods as well. This underscores the pivotal role of TSDs in optimizing model behavior for downstream tasks, highlighting their universal applicability across various PEFT strategies.

Table 5: Results of commonsense reasoning tasks when fine-tuning LLaMA-7B.

| Method | # Params (%) | BoolQ | PIQA | SIQA | HellaSwag | WinoGrande | ARC-e | ARC-c | OBQA | Avg. |
|---|---|---|---|---|---|---|---|---|---|---|
| ChatGPT | - | 73.1 | 85.4 | 68.5 | 78.5 | 66.1 | 89.8 | 79.9 | 74.8 | 77.0 |
| AdaLoRA$_{r=32}$ | 0.83 | 69.1 | 82.2 | 77.2 | 78.3 | 78.2 | 79.7 | 61.9 | 77.2 | 75.5 |
| AdaLoRA-Dash | 0.83 | 69.4 | 82.1 | 77.5 | 77.9 | 75.1 | 80.3 | 63.8 | 79.8 | **75.7** |
| FLoRA$_{r=32}$ | 0.83 | 66.4 | 81.3 | 77.1 | 75.6 | 77.1 | 77.2 | 62.4 | 77.6 | 74.3 |
| FLoRA-Dash | 0.83 | 69.8 | 81.9 | 78.0 | 83.3 | 79.6 | 79.1 | 62.7 | 79.4 | **76.7** |

## 7 CONCLUSION

In this paper, we revisit LoRA's exploration of task-specific directions (TSDs), highlighting its inadequacies in understanding TSDs. We then provide a precise definition of TSD and delve into their properties to better understand their role in fine-tuning large language models. Building on this foundational knowledge, we introduce LoRA-Dash, designed to fully unleash the potential of TSDs. Through comprehensive experiments and in-depth analysis, we demonstrate the significant advantages of LoRA-Dash over conventional methods. Our findings not only validate the effectiveness of LoRA-Dash but also highlight the crucial importance of TSDs in achieving superior task-specific performance. By pushing the boundaries of parameter-efficient fine-tuning, we aim to inspire continued research and development in this vibrant area, transforming practices across diverse applications in natural language processing and beyond.

ACKNOWLEDGEMENTS

This work was supported by NSFC 62322604, NSFC 62176159, and Shanghai Municipal Science and Technology Major Project 2021SHZDZX0102.

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

# Contents

## A  STRUCTURE AND CONTRIBUTIONS OF THIS PAPER

**The structure of this paper is as follows**:

- We begin by reviewing the description and exploration of TSDs within the LoRA framework. Our analysis reveals the significance of the clear definition of TSDs within PEFT (Sec. 2).
- We propose a novel framework that clearly defines what TSDs are (Sec. 3.2) and further explores their properties (Sec. 3.3). Additionally, we highlight the challenges of applying TSDs in fine-tuning processes (Sec. 3.4).
- To further unleash the potential of TSDs, we propose a new method, LoRA-Dash, which leverages TSDs to enhance performance on downstream tasks (Sec. 4). We first introduce how LoRA-Dash identifies TSDs (Sec. 4.1), followed by an explanation of how it leverages this information (Sec. 4.2).
- We conduct various experiments to further analyse the underlying mechanisms of LoRA-Dash (Sec. 5-6).

**The contributions of this paper are as follows**:

- We propose a framework, which clearly defines TSDs, and offers foundations for further analysis.
- We propose a novel PEFT method LoRA-Dash, which identified TSDs and leverage the potential of them to enhance performance on downstream tasks.

## B  DETAILS ON TASK-SPECIFIC DIRECTIONS

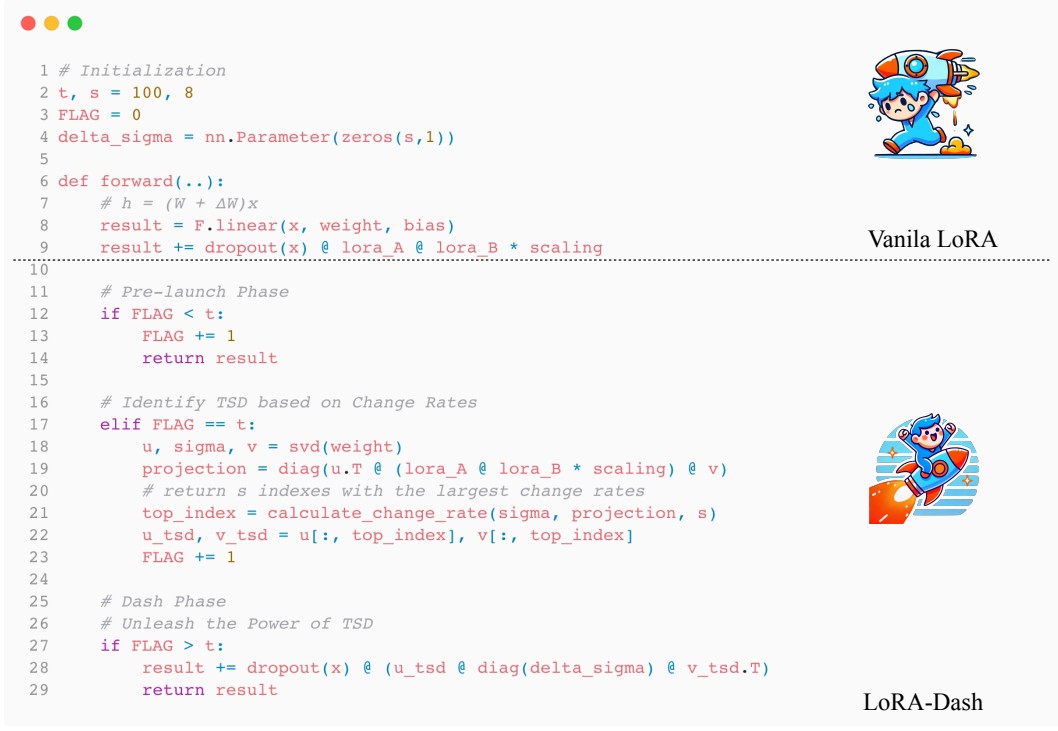

```
1  # Initialization
2  t, s = 100, 8
3  FLAG = 0
4  delta_sigma = nn.Parameter(zeros(s,1))
5
6  def forward(..):
7      # h = (W + ΔW)x
8      result = F.linear(x, weight, bias)
9      result += dropout(x) @ lora_A @ lora_B * scaling
```
Vanila LoRA

```
10
11     # Pre-launch Phase
12     if FLAG < t:
13         FLAG += 1
14         return result
15
16     # Identify TSD based on Change Rates
17     elif FLAG == t:
18         u, sigma, v = svd(weight)
19         projection = diag(u.T @ (lora_A @ lora_B * scaling) @ v)
20         # return s indexes with the largest change rates
21         top_index = calculate_change_rate(sigma, projection, s)
22         u_tsd, v_tsd = u[:, top_index], v[:, top_index]
23         FLAG += 1
24
25     # Dash Phase
26     # Unleash the Power of TSD
27     if FLAG > t:
28         result += dropout(x) @ (u_tsd @ diag(delta_sigma) @ v_tsd.T)
29         return result
```
LoRA-Dash

Figure 6: Pseudo codes of LoRA-Dash. LoRA is capable of identifying TSDs, but it does not utilize the corresponding information. LoRA-Dash builds on this by actively leveraging the power of these identified TSDs to enhance model performance significantly.

### B.1 How Do Task-specific Directions Arise in LoRA?

To explore the relationship between the learned $\Delta\mathbf{W}$ and the original weights $\mathbf{W}$, LoRA initially applies SVD to $\Delta\mathbf{W}$ to extract its left and right singular vectors, $\mathbf{U}$ and $\mathbf{V}$. ***LoRA first validates that the directions corresponding to the top singular vectors of*** $\Delta\mathbf{W}$ ***tend to overlap significantly across different ranks*** (conclusion 1). Subsequently, LoRA projects $\mathbf{W}$ onto the $r$-dimensional subspace defined by $\Delta\mathbf{W}$, calculating $\mathbf{U}^\mathsf{T}\mathbf{W}\mathbf{V}$ and its Frobenius norm $\|\mathbf{U}^\mathsf{T}\mathbf{W}\mathbf{V}\|_F$. This norm is also computed by replacing the top $r$ singular vectors $\mathbf{U}$ and $\mathbf{V}$ derived from $\mathbf{W}$ or a random matrix for comparison. The results reveal that $\|\mathbf{U}^\mathsf{T}\mathbf{W}\mathbf{V}\|_F$ when $\mathbf{U}$ and $\mathbf{V}$ are top singular vectors derived from $\Delta\mathbf{W}$ or $\mathbf{W}$ is much greater than when they are derived from a random matrix, suggesting a stronger correlation of $\Delta\mathbf{W}$ with $\mathbf{W}$ than with a random matrix. ***This indicates that the features*** $\Delta\mathbf{W}$ ***amplifies already present in*** $\mathbf{W}$.

Additionally, LoRA introduces the "feature amplification factor" to measure the extent of feature enhancement, defined as $\|\Delta\mathbf{W}\|_F/\|\mathbf{U}^\mathsf{T}\mathbf{W}\mathbf{V}\|_F$. The factor is significantly higher when $\mathbf{U}$ and $\mathbf{V}$ are top singular vectors derived from $\Delta\mathbf{W}$ compared to when they are those derived from $\mathbf{W}$, ***suggesting that*** $\Delta\mathbf{W}$ ***only boosts directions that are not emphasized in*** $\mathbf{W}$. Moreover, a larger $r$ ($r = 64$) yields a much lower amplification factor than a smaller $r$ ($r = 4$), ***implying that the number of "task-specific directions" is small***. Through extensive experiments across various ranks and layers, LoRA determines that $\Delta\mathbf{W}$ potentially amplifies important directions for specific downstream tasks that were learned but not emphasized during general pre-training, and refers to them as "task-specific" directions (TSDs).

LoRA has drawn three conclusions related to TSDs:

1. TSDs are the directions of top singular vectors derived from the learned $\Delta\mathbf{W} = \mathbf{AB}$. TSDs consistently exhibit significant overlap across different $r$ settings used in the learning of $\Delta\mathbf{W}$, where $r$ is the rank configuration of $\Delta\mathbf{W}$.

2. TSDs are not the directions of top singular vectors derived from the pretrained weights $\mathbf{W}$.

3. TSDs are certain directions that have already been learned by $\mathbf{W}$ but were not emphasized.

However, only the first conclusion of LoRA already presents significant contradictions. For a specific task, TSDs should indeed remain consistent and not depend on learned $\Delta\mathbf{W}$. Moreover, revisiting the first and the third conclusions, we could deduce that the top singular directions of $\Delta\mathbf{W}$ are among the directions of $\mathbf{W}$. Nonetheless, in practice, the singular directions of $\Delta\mathbf{W}$ are unlikely to align exactly with any specific directions in $\mathbf{W}$. As shown in our paper, although LoRA highlights the importance of TSDs for downstream tasks, it struggles with a clear definition of TSDs which leads to contradictions, and does not effectively outline how they can be leveraged, leading to potential under-utilization in practical applications

For more details on what they did and why they drew the conclusions, we encourage readers to refer to their original paper Hu et al. (2021), specifically Section 7.

### B.2 Why should TSD be Established on $\mathbf{W}$?

Indeed, the most accurate direction for TSD should be the direction of $\Delta\mathbf{W}^*$. However, since we do not have access to $\Delta\mathbf{W}^*$ or $\mathbf{W}^*$ during fine-tuning, we only have information about $\mathbf{W}$. Therefore, we establish TSD within the coordinate system based on the directions of $\mathbf{W}$.

### B.3 Why Must TSD be in the Core Bases Rather Than Other Global Bases?

Task-specific directions are defined as a subset of the core bases. However, why can't they be defined as a set of bases from the global bases of $\mathbf{W}$ instead? Indeed, other global bases might also experience significant coordinate changes. However, the primary reason for not considering these bases is the difficulty in defining them.

Firstly, if TSDs were defined based on the change rates, since the coordinate values of $\mathbf{W}$ on the other global bases are zero, even a minimal projection of $\Delta\mathbf{W}$ onto these global bases could result in an excessively high change rate, which would be unreasonable. Secondly, if TSDs were defined based on the absolute magnitude of coordinate changes, consider a scenario where a coordinate

value in the core basis and a coordinate value in the global basis change by the same amount. If the original coordinate value in the core basis was significantly larger than the change, the change, even if substantial, could be viewed as a minor perturbation relative to the original value. Therefore, the same magnitude of change does not necessarily imply that the corresponding directions are equally important. For these reasons, we define TSDs based solely on the rate of change and restrict their definition to the core bases.

## C DETAILS ON LoRA-DASH

### C.1 ALGORITHM OF LoRA-DASH

The algorithm of LoRA-Dash is shown in Fig. 6.

### C.2 MATRIX OPERATION OF DASH PHASE

Since in Sec. 4.2, we have adopted vector calculations for change rates for clear representation. However, to enhance efficiency, we will transition to matrix computations for determining change rates.

Let $\mathbf{U} = [\mathbf{u}_1 \quad \ldots \quad \mathbf{u}_n] \in \mathbb{R}^{n \times n}$ and $\mathbf{V} = [\mathbf{v}_1 \quad \ldots \quad \mathbf{v}_m] \in \mathbb{R}^{m \times m}$ represent the left and right singular vectors for a weight matrix $\mathbf{W} \in \mathbb{R}^{n \times m}$ $(n < m)$, we have

$$\mathbf{U}^\mathsf{T} \Delta \mathbf{W} \mathbf{V} = \begin{bmatrix} \mathbf{u}_1^\mathsf{T} \\ \vdots \\ \mathbf{u}_n^\mathsf{T} \end{bmatrix} \Delta \mathbf{W} \begin{bmatrix} \mathbf{v}_1 & \cdots & \mathbf{v}_m \end{bmatrix} = \begin{bmatrix} \mathbf{u}_1^\mathsf{T} \Delta \mathbf{W} \mathbf{v}_1 & \ldots & \mathbf{u}_1^\mathsf{T} \Delta \mathbf{W} \mathbf{v}_m \\ \vdots & \mathbf{u}_i^\mathsf{T} \Delta \mathbf{W} \mathbf{v}_j & \vdots \\ \mathbf{u}_n^\mathsf{T} \Delta \mathbf{W} \mathbf{v}_1 & \ldots & \mathbf{u}_n^\mathsf{T} \Delta \mathbf{W} \mathbf{v}_m \end{bmatrix}. \quad (7)$$

Therefore, we can derive the change rates $\delta = [\delta_1 \quad \ldots \quad \delta_n]^\mathsf{T}$, which are the diagonal elements of the matrix $\mathbf{U}^\mathsf{T} \Delta \mathbf{W} \mathbf{V}$.

Following a similar process, we can rewrite Eq. (5) as

$$\sum_{i=1}^{s} \Delta \sigma_i \bar{\mathbf{u}}_i \bar{\mathbf{v}}_i^\mathsf{T} = \bar{\mathbf{U}} \Delta \mathbf{\Sigma} \bar{\mathbf{V}}^\mathsf{T}, \quad (8)$$

where $\bar{\mathbf{U}} = [\bar{\mathbf{u}}_1 \quad \ldots \quad \bar{\mathbf{u}}_s]$, $\bar{\mathbf{V}} = [\bar{\mathbf{v}}_1 \quad \ldots \quad \bar{\mathbf{v}}_s]$ and $\Delta \mathbf{\Sigma} = \mathrm{diag}([\Delta \sigma_1 \quad \ldots \quad \Delta \sigma_s])$.

## D DETAILS ON EXPERIMENTS

### D.1 IMPLEMENTATION DETAILS

We compare LoRA-Dash mainly with LoRA. The ranks of LoRA-Dash, LoRA and other methods are varied among {2, 4, 8, 16, 32, 64}. All methods are implemented using the publicly available PyTorch Paszke et al. (2019) framework, and all experiments are conducted on NVIDIA A100 GPUs[4]. The hyper-parameter $t$ of LoRA-Dash is set to 100, and $s = 8$. We fine-tune all the layers of different models in all the experiments, and the detailed experimental settings are shown in the following Secs. D.4-D.6.

### D.2 DETAILS ON FULLY FINE-TUNING LLaMA IN SECTION 3.3

To further explore the properties of TSDs, we fully fine-tune LLaMA-7B Touvron et al. (2023a) on commonsense reasoning tasks. For the commonsense reasoning tasks, which consist of 8 distinct sub-tasks, each defined with specific training and testing sets, we have adopted the approach in Hu et al. (2023). Following this method, we amalgamated the training datasets from all sub-tasks to construct a comprehensive final training dataset. More details can be found in Sec. D.4. The experiment was conducted on 8 A100 GPUs.

---

[4]We have also test that most of the experiments can also be conducted on one consumer GPU resource such as NVIDIA RTX3090.

The change rates in Fig. 1 are derived from the weights obtained from the self-attention value projection in the 16th layer. However, our validation confirms that the results and conclusions drawn from the diagrams are consistent, regardless of which layer or specific module within the attention mechanism is analyzed. For the original change rate $\delta$ for a core basis, it is scaled as $\ln(\delta + 1)/3$.

Additionally, it is worth noting that the phenomenon presented in Fig. 1 is not an isolated case. Similar behavior has been observed in other tasks and models. For instance, as shown in Fig., results obtained by training Qwen2.5-7B Team (2024) on the math reasoning task Hu et al. (2023) exhibit patterns that are fundamentally consistent with our observations in Fig. 1.

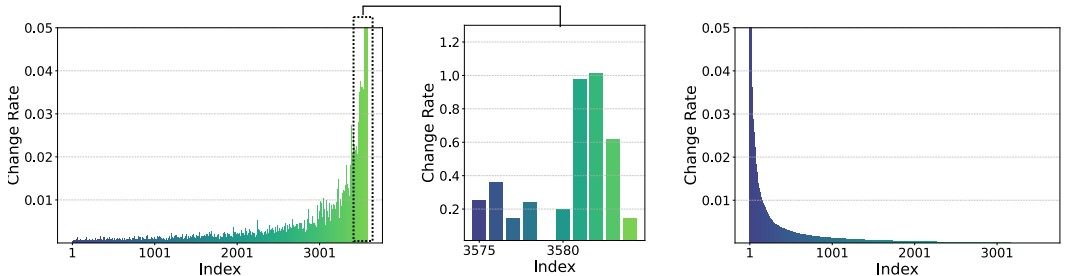

Figure 7: The results when fine-tuning Qwen2.5-7B on math reasoning task are similar to those in Fig. 1

### D.3 Details on Fine-tuning LLaMA with LoRA in Section 3.4

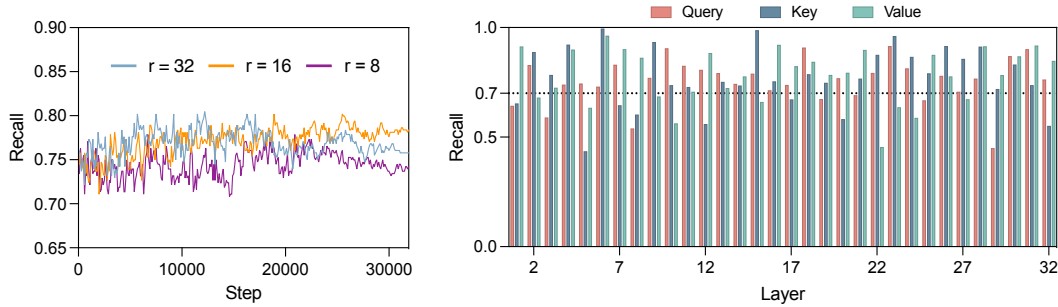

Figure 8: We track the recall of TSDs every 100 training steps in the query, key and value layers of the LLaMA-7B during LoRA fine-tuning, analyzing how well the continuous updated $\Delta \mathbf{W}$ captures TSDs. The left and right figure settings are the same as those in Fig. 2.

We fine-tune LLaMA-7B using LoRA under three ranks, and for every 100 training steps we record the top 8 core directions of $\mathbf{W}$ which exhibit the highest change rates based on the projection of continuously updated $\Delta \mathbf{W}$ (i.e., the directions related to top 8 largest $\delta_i = \mathbf{u}_i^\mathsf{T} \Delta \mathbf{W} \mathbf{v}_i / \sigma_i$, where $\mathbf{u}_i$, $\mathbf{v}_i$ and $\sigma_i$ are the corresponding left/right singular vectors and values of $\mathbf{W}$). These directions serve as the "TSDs" predicted by LoRA. We denote these directions as "launched TSDs" (LTSDs) for differentiation. We then identify the top 4 or 16 TSDs with the highest change rates in Sec. 3.3.

We conduct two experimental setups to explore that:

1. How many of the top 4 TSDs are contained in the 8 LTSDs;
2. How many LTSDs are contained in the top 16 TSDs.

The purpose of the first experiment is more intuitive, aiming to confirm whether the TSDs with the *top highest change rates* can indeed be captured by $\Delta \mathbf{W}$ from LoRA. The second experiment, which may seem a bit more obscure, is designed to address concerns that, besides capturing the top-ranked

TSDs, the LTSDs identified might include irrelevant directions. Therefore, this second setup allows us to examine whether the LTSDs predominantly fall within the upper echelons of TSDs based on their change rates, providing insight into the overall relevance and accuracy of the LTSDs identified by LoRA. Specifically, we employ two metrics to quantify the quality of LTSDs:

- **Accuracy**: Measures how many of the 8 LTSDs are correctly included in the top 16 TSDs.
- **Recall**: Assesses how many of the top 4 TSDs are captured among the 8 LTSDs.

The results presented in Figs 2 and 8 lead to two significant observations regarding the effectiveness of using $\Delta \mathbf{W}$ to identify TSDs:

- **Precision in Identifying Crucial TSDs**: The LTSDs captured by $\Delta \mathbf{W}$ include several of the actual TSDs with the highest change rates, demonstrating that the predicted directions are not only accurate but pinpoint some of the most critical TSDs.

- **Overall Coverage of Significant TSDs**: The entirety of the LTSDs resides within the upper echelon of TSDs in terms of change rates. This suggests that while the primary directions identified are indeed the most vital, the remaining directions predicted also belong to the upper tier of important TSDs.

These insights affirm the utility of $\Delta \mathbf{W}$ in accurately capturing and representing the information of real TSDs. The method shows potential in leveraging learned directional changes to identify and utilize key features specific to given tasks, underscoring the robustness of $\Delta \mathbf{W}$ as a predictive tool in practical applications.

For implementation details, please refer to Sec. D.4.

### D.4    DETAILS ON COMMONSENSE REASONING TASK

The commonsense reasoning benchmarks consist of 8 distinct sub-tasks, each with its designated dataset, i.e., BoolQ Clark et al. (2019), PIQA Bisk et al. (2020), SIQA Sap et al. (2019), HellaS. Zellers et al. (2019), WinoG. Sakaguchi et al. (2021), ARC-e/ARC-c Clark et al. (2018), OBQA Mihaylov et al. (2018). Adhering to the protocol outlined by Hu et al. (2023), we merge the training datasets from all tasks to form a comprehensive training dataset (Commonsense170K dataset), subsequently performing evaluations against each individual task's testing set. The template to create commonsense reasoning task is shown in Table. 6.

We fine-tune LLaMA-7B Touvron et al. (2023a), LLaMA2-7B Touvron et al. (2023b) and LLaMA3-8B AI@Meta (2024) on this task. Additionally, we integrate results from ChatGPT's implementation with the gpt-3.5-turbo API, particularly focusing on zero-shot Chain of Thought approaches Wei et al. (2022).

To ensure equitable comparisons, initial fine-tuning for models utilizing LoRA-Dash is conducted under the LoRA configurations while only varying the learning rate to optimize performance. The hyper-parameter settings of LoRA-Dash are shown in Table. 7. The results of LoRA are cited from Hu et al. (2023); Liu et al. (2024).

We also represent some answer samples with fine-tuned LLaMA-7B. The answers are shown in Tables. 8-15.

### D.5    DETAILS ON NATURAL LANGUAGE UNDERSTANDING TASK

For natural language understanding (NLU) task, we adopt the General Language Understanding Evaluation (GLUE) Wang et al. (2018) benchmark, which is designed to test capabilities across various tasks. This benchmark consists of two single-sentence classification tasks, CoLA Warstadt et al. (2019) and SST-2 Socher et al. (2013), three similarity and paraphrase tasks, MRPC Dolan & Brockett (2005), QQP Wang et al. (2018), and STS-B Cer et al. (2017), and three natural language inference tasks, MNLI Williams et al. (2017), QNLI Rajpurkar et al. (2016), and RTE Dagan et al. (2005); Bar-Haim et al. (2006); Giampiccolo et al. (2007); Bentivogli et al. (2009). The details of these datasets are shown in Table. 16.

Table 6: The template of each dataset used to create commonsense reasoning data for fine-tuning.

| Dataset | Fine-tuning Data Template |
|---------|---------------------------|
| BoolQ | Please answer the following question with true or false, question: [QUESTION]
Answer format: true/false
the correct answer is [ANSWER] |
| PIQA | Please choose the correct solution to the question: [QUESTION]
Solution1: [SOLUTION_1]
Solution2: [SOLUTION_2]
Answer format: solution1/solution2
the correct answer is [ANSWER] |
| SIQA | Please choose the correct solution to the question: [QUESTION]
Answer1: [ANSWER_1]
Answer2: [ANSWER_2]
Answer3: [ANSWER_3]
Answer format: answer1/answer2/answer3
the correct answer is [ANSWER] |
| HellaSwag | Please choose the correct ending to complete the given sentence: [ACTIVITY_LABEL]: [CONTEXT]
Ending1: [ENDING_1]
Ending2: [ENDING_2]
Ending3: [ENDING_3]
Ending4: [ENDING_4]
Answer format: ending1/ending2/ending3/ending4
the correct answer is [ANSWER] |
| WinoGrande | Please choose the correct answer to fill in the blank to complete the given sentence: [SENTENCE]
Option1: [OPTION_1]
Option2: [OPTION_2]
the correct answer is [ANSWER] |
| ARC-e & ARC-c | Please choose the correct solution to the question: [QUESTION]
Answer1: [ANSWER_1]
Answer2: [ANSWER_2]
Answer3: [ANSWER_3]
Answer4: [ANSWER_4]
Answer format: answer1/answer2/answer3/answer4
the correct answer is [ANSWER] |
| OBQA | Please choose the correct solution to the question: [QUESTION]
Answer1: [ANSWER_1]
Answer2: [ANSWER_2]
Answer3: [ANSWER_3]
Answer4: [ANSWER_4]
Answer format: answer1/answer2/answer3/answer4
the correct answer is [ANSWER] |

Table 7: Hyper-parameter settings of LoRA-Dash on commonsense reasoning task.

| Hyper-parameters | LLaMA-7B | | | | | LLaMA2-7B | | LLaMA3-8B | |
|------------------|------|------|------|------|--------|------|------|------|--------|
| Rank $r$ | 4 | 8 | 16 | 32 | 64 | 16 | 32 | 16 | 32 |
| $\alpha$ | 8 | 16 | 32 | 64 | 128 | 32 | 64 | 32 | 64 |
| LR | 5e-4 | 4e-4 | 5e-4 | 1e-4 | 0.9e-4 | 2e-4 | 1e-4 | 2e-4 | 0.8e-4 |
| LR Scheduler | Linear | | | | | | | | |
| Dropout | 0.05 | | | | | | | | |
| Optimizer | AdamW | | | | | | | | |
| Batch size | 16 | | | | | | | | |
| Warmup Steps | 100 | | | | | | | | |
| Epochs | 3 | | | | | | | | |
| Where | Q, K, V, Up, Down | | | | | | | | |

Table 8: An example from BoolQ. The outputs of LLaMA-7B with LoRA and LoRA-Dash.

| Question | was whiskey galore based on a true story? |
|---|---|
| Label | true |
| LoRA
LoRA-Dash | the correct answer is false
the correct answer is true |

Table 9: An example from PIQA. The outputs of LLaMA-7B with LoRA and LoRA-Dash.

| Question | To add chocolate and caramel to hot milk and cream.
**Solution1**: Chop the chocolate into small chunks.
Bring the milk and cream to a simmer on a low-medium heat.
Once the mixture begins to cool, remove from the heat and add the chocolate and caramel sauce.
**Solution2**: Chop the chocolate into small chunks.
Bring the milk and cream to a simmer on a low - medium heat.
Once the mixture begins to simmer, remove from the heat and add the chocolate and caramel sauce. |
|---|---|
| Label | solution2 |
| LoRA
LoRA-Dash | the correct answer is solution1
the correct answer is solution2 |

Table 10: An example from SIQA. The outputs of LLaMA-7B with LoRA and LoRA-Dash.

| Question | Lee challenged her to a fight after she talked bad about Lee to his friends.
How would Others feel as a result?
**Answer1**: happy that Lee wants a fight
**Answer2**: angry
**Answer3**: fight |
|---|---|
| Label | answer2 |
| LoRA
LoRA-Dash | the correct answer is answer1
the correct answer is answer2 |

Table 11: An example from HellaSwag. The outputs of LLaMA-7B with LoRA and LoRA-Dash.

| Question | Personal Care and Style: [header] How to dress rugged (men) [title] Shop for pants.
[step] Great rugged outfits start with the right pants.
Shop for jeans, khakis and canvas with straight-cut legs, and avoid any flashy graphics or loud colors.
**Ending1**: Also, make sure you try on pants to ensure they fit comfortably.
Rugged styles are about utility, so the pants shouldn't be extremely tight or extremely baggy:
look for "relaxed fit" or "straight-cut".
**Ending2**: Men should try slacks, dress pants, or street clothes for a rugged look.
[substeps] Choose gut-length, flared, or boot-cut trousers.
**Ending3**: A good rule of thumb is to dress in the right pair of pants, and balance it with that.
Consider clothing tucked into your upper thighs with high-rise, thigh high, or flare pants.
**Ending4**: Lightweight and comfortable feet are always a major consideration when shopping for a rugged shirt.
[substeps] These hiking boots, while lightweight and appropriate for longer, are a must. |
|---|---|
| Label | ending1 |
| LoRA
LoRA-Dash | the correct answer is ending2
the correct answer is ending1 |

We fine-tune DeBERTaV3-base and DeBERTaV3-large He et al. (2021b) models on this task. The hyper-parameter settings for this task is shown in Table. 17.

Table 12: An example from WinoGrande. The outputs of LLaMA-7B with LoRA and LoRA-Dash.

| Question | Lawrence was always gaining weight while Dennis was losing it as _ liked to eat too little.
**Option1**: Lawrence
**Option2**: Dennis |
|---|---|
| Label | option2 |
| LoRA
LoRA-Dash | the correct answer is option1
the correct answer is option2 |

Table 13: An example from ARC-e. The outputs of LLaMA-7B with LoRA and LoRA-Dash.

| Question | There are a variety of events that can disrupt ecosystems.
Some events result in local disturbances and others can affect a much larger area.
Which event would most likely result in change on a global scale?
**Answer1**: land scarred by forest fires
**Answer2**: erosion of ocean shorelines
**Answer3**: converging plate boundaries
**Answer4**: destruction of tropical rain forests |
|---|---|
| Label | answer4 |
| LoRA
LoRA-Dash | the correct answer is answer3
the correct answer is answer4 |

Table 14: An example from ARC-c. The outputs of LLaMA-7B with LoRA and LoRA-Dash.

| Question | What is a similarity between sound waves and light waves?
**Answer1**: Both carry energy.
**Answer2**: Both travel in vacuums.
**Answer3**: Both are caused by vibrations.
**Answer4**: Both are traveling at the same speed. |
|---|---|
| Label | answer1 |
| LoRA
LoRA-Dash | the correct answer is answer3
the correct answer is answer1 |

Table 15: An example from OBQA. The outputs of LLaMA-7B with LoRA and LoRA-Dash.

| Question | Which best demonstrates the concept of force causing an increase in speed?
**Answer1**: skating on a rough surface
**Answer2**: a full bag swung in circles
**Answer3**: a computer powering on
**Answer4**: a baker stirring batter |
|---|---|
| Label | answer2 |
| LoRA
LoRA-Dash | the correct answer is answer1
the correct answer is answer2 |

## D.6 DETAILS ON SUBJECT-DRIVEN GENERATION TASK

In our experiment, we fine-tune the text-to-image diffusion models specifically tailored for subject-driven generation tasks, as outlined in recent research Ruiz et al. (2023). The objective of this task is to generate images that adhere closely to prompts associated with a particular subject, defined by a few exemplar images. This involves initially fine-tuning a text-to-image model using image-text pairs where the text contains a unique identifier (e.g., "A photo of a [V] cat"). Subsequent

Table 16: Details of GLUE dataset.

| Dataset | Task | # Train | # Dev | # Test | # Label | Metrics |
|---------|------|---------|-------|--------|---------|---------|
| Single-Sentence Classification | | | | | | |
| CoLA | Acceptability | 8.5k | 1k | 1k | 2 | Matthews corr |
| SST-2 | Sentiment | 67k | 872 | 1.8k | 2 | Accuracy |
| Similarity and Paraphrase | | | | | | |
| MRPC | Paraphrase | 3.7k | 408 | 1.7k | 2 | Accuracy / F1 |
| QQP | Paraphrase | 364k | 40k | 391k | 2 | Accuracy / F1 |
| STS-B | Similarity | 7k | 1.5k | 1.4k | 1 | Pearson/ Spearman Corr |
| Natural Language Inference | | | | | | |
| MNLI | NLI | 393k | 20k | 20k | 3 | Accuracy |
| QNLI | QA/NLI | 108k | 5.7k | 5.7k | 2 | Accuracy |
| RTE | NLI | 2.5k | 276 | 3k | 2 | Accuracy |

Table 17: Hyper-parameter settings of LoRA-Dash on NLU task.

| Hyper-parameter | MNLI | SST-2 | CoLA | QQP | QNLI | RTE | MRPC | STS-B |
|-----------------|------|-------|------|-----|------|-----|------|-------|
| Optimizer | AdamW | | | | | | | |
| Warmup Ratio | 0.1 | | | | | | | |
| LR schedule | Linear | | | | | | | |
| Rank $r$ | 2 & 8 | | | | | | | |
| LoRA alpha | 4 & 16 | | | | | | | |
| Max Seq. Len. | 256 | 128 | 64 | 320 | 512 | 320 | 320 | 128 |
| Batch Size | 32 | 32 | 32 | 32 | 32 | 32 | 32 | 32 |
| Learning Rate | 5e-4 | 8e-4 | 8e-4 | 1e-3 | 5e-4 | 1.2e-3 | 1e-3 | 5e-4 |
| Epochs | 12 | 24 | 25 | 5 | 5 | 50 | 30 | 25 |

image generations are driven by new prompts incorporating this identifier, aiming to produce images aligned with the learned subject.

For this experiment, we use the SDXL5 model Podell et al. (2023), applying both LoRA and LoRA-Dash methods for fine-tuning. The fine-tuning process is conducted with a learning rate of 1e-4 and a batch size of 4. We train the model over 500 steps on a single 80GB A100 GPU, taking approximately 23 minutes to complete. For the generation phase, we execute 50 inference steps for each given prompt to synthesize the final images, which takes approximately 7 seconds to complete.

We mainly adopt the official data of DreamBooth Ruiz et al. (2023) for diffusion.

## D.7 DETAILS ON THE AMPLIFICATION OF TSD IN SEC. 6.2

To explore to what extent LoRA-Dash can amplify the information of the TSDs it determines, we project the pretrained weights $\mathbf{W}$ and the weights merged after training, i.e., $\mathbf{W} + \Delta\mathbf{W}_{all}$, onto the LTSDs. Let $\bar{\mathbf{U}} = [\bar{\mathbf{u}}_1 \quad \dots \quad \bar{\mathbf{u}}_s]$ and $\bar{\mathbf{V}} = [\bar{\mathbf{v}}_1 \quad \dots \quad \bar{\mathbf{v}}_s]$, we can then define the amplification factor of $\Delta\mathbf{W}_{all}$ as the Frobenius norms of these two weights on these directions, which is given

by:

$$\frac{\|\bar{\mathbf{U}}^\mathsf{T}(\mathbf{W} + \Delta\mathbf{W}_{all})\bar{\mathbf{V}}\|_F}{\|\bar{\mathbf{U}}^\mathsf{T}\mathbf{W}\bar{\mathbf{V}}\|_F}. \tag{9}$$

Specifically, we also evaluate the separate impacts of $\Delta\mathbf{W}_{AB}$ and $\Delta\mathbf{W}_{dash}$ in the amplification, with the corresponding amplification factor being

$$\frac{\|\bar{\mathbf{U}}^\mathsf{T}(\mathbf{W} + \Delta\mathbf{W}_{AB})\bar{\mathbf{V}}\|_F}{\|\bar{\mathbf{U}}^\mathsf{T}\mathbf{W}\bar{\mathbf{V}}\|_F} \quad \text{and} \quad \frac{\|\bar{\mathbf{U}}^\mathsf{T}(\mathbf{W} + \Delta\mathbf{W}_{dash})\bar{\mathbf{V}}\|_F}{\|\bar{\mathbf{U}}^\mathsf{T}\mathbf{W}\bar{\mathbf{V}}\|_F}. \tag{10}$$

The results of the amplification are shown in Table. 4.

Moreover, as shown in Sec. B.2, TSDs are indeed the projection of $\Delta\mathbf{W}^*$'s direction onto the core bases of $\mathbf{W}$. Therefore, we also explored the amplification factor along the direction of $\Delta\mathbf{W}_{all}$. We replaced $\bar{\mathbf{U}}$ and $\bar{\mathbf{V}}$ in the previous amplification factor calculation with the top $r$ left and right singular vectors of $\Delta\mathbf{W}_{all}$, where $r$ is the rank setting of LoRA-Dash. We reported the corresponding results in Table. 18, where we can draw a conclusion:

Table 18: Amplification of $\Delta\mathbf{W}_{all}$ of LoRA-Dash onto TSDs or $\Delta\mathbf{W}_{all}$'s direction. We report the amplification factor from the self-attention key (k) projection of the first, middle and last layer of LLaMA-7B, as well as the average of all q, k, and v modules.

| **U** and **V** from | $r = 8$ | | $r = 16$ | | $r = 32$ | |
|---|---|---|---|---|---|---|
| | TSD | $\Delta\mathbf{W}_{all}$ | TSD | $\Delta\mathbf{W}_{all}$ | TSD | $\Delta\mathbf{W}_{all}$ |
| 1st | 282.15 | 22.75 | 628.60 | 30.13 | 42.00 | 3.07 |
| 16th | 11.15 | 144.89 | 13.07 | 119.18 | 2.16 | 9.92 |
| 32nd | 68.02 | 75.81 | 30.87 | 120.76 | 8.19 | 10.63 |
| Avg. | 22.85 | 153.49 | 32.12 | 126.26 | 5.28 | 11.31 |

- The amplification factor on the directions of $\Delta\mathbf{W}_{all}$ is significantly higher than that on the TSDs. Since TSDs can be considered as a projection of $\Delta\mathbf{W}$ onto the coordinate system of $\mathbf{W}$, it is natural that some information is lost in this projection for we only select $s$ TSDs, leading to a lower amplification factor on the TSD compared to before the projection.

Additionally, we observed an interesting trend: the amplification factor on $\Delta\mathbf{W}_{all}$'s directions significantly decreases as the rank increases. In LoRA Hu et al. (2021), the authors claimed that the occurrence of a small number of TSDs leads to this situation. However, a more plausible explanation is that higher rank corresponds to $\Delta\mathbf{W}$ containing more irrelevant directions, thereby reducing the amplification factor.

To validate this, consider the setting of $r = 32$ of $\Delta\mathbf{W}_{all}$. The 32 directions of $\Delta\mathbf{W}_{all}$ include some irrelevant directions, which is why the amplification factor is lower. This implies that if the irrelevant directions among these 32 are removed, the amplification factor could potentially increase. To explore whether this is the case, we recalculated the amplification factors using the top 2, 4, 8 and 16 singular directions of $\Delta\mathbf{W}_{all}$, and the results are shown in Table. 19. The observation that the amplification factor decreases as projections are made onto an increasing number of the top singular directions of $\Delta\mathbf{W}_{all}$ indicates that $\Delta\mathbf{W}_{all}$'s singular directions do indeed contain irrelevant directions in its bottom singular directions. It is not surprising, since TSDs are only a few directions, as shown in its properties. Therefore, it also motivates us to select several top directions in prelaunch phase of LoRA-Dash.

However, we believe that there is no necessary correlation between amplification factors and the effectiveness of a model. Amplification factors simply reflect how much the model amplifies features in certain directions, and the specific degree to which this should occur is unknown.

Table 19: Amplification of $\Delta\mathbf{W}_{all}$ of LoRA-Dash onto different number of top singular directions of $\Delta\mathbf{W}_{all}$. We report the amplification factor from the self-attention key (k) projection of the first, middle and last layer of LLaMA-7B, as well as the average of all q, k, and v modules.

| $\mathbf{U}$ and $\mathbf{V}$ are directions of top | 2 | 4 | 8 | 16 | 32 |
|---|---|---|---|---|---|
| 1st | 11.93 | 8.65 | 6.44 | 4.34 | 3.07 |
| 16th | 71.82 | 50.09 | 31.98 | 17.89 | 9.92 |
| 32nd | 160.18 | 58.17 | 30.82 | 19.15 | 10.63 |
| Avg. | 139.03 | 70.42 | 38.87 | 21.02 | 11.31 |

# E    MORE EXPERIMENT RESULTS

## E.1    MORE RESULTS FOR SUBJECT-DRIVEN GENERATION TASK

We here provide more visual results for the subject-driven generation task in Fig. 9.

We also conducted a user study using the DreamBooth dataset. We selected 10 different categories and generated corresponding images using five prompts for each category. We then invited 36 experts to evaluate the images based on subject similarity and prompt consistency, with each expert evaluating randomly sampled 25 pairs of images. During evaluation, the pairs of images were shuffled, and the experts were unaware of the source model for each image. Overall, the images LoRA-Dash generated achieved an 84.51% approval rating. This suggests that the images generated by LoRA-Dash were more favorably recognized by experts in terms of subject similarity and prompt consistency.

We also evaluated the performance using the text-image score evaluated by CLIP Radford et al. (2021) and the image similarity score. The CLIP text-image score measures the alignment between textual descriptions and the corresponding images, while the image similarity score quantifies the similarity between the original image and the generated image. The results are presented in Table 20, where it is evident that LoRA-Dash outperforms LoRA on both metrics. The images generated by LoRA-Dash exhibit better text alignment and a stronger correspondence with the original images, demonstrating that LoRA-Dash achieves superior results compared to LoRA.

Table 20: Performance comparison between LoRA and LoRA-Dash on CLIP text-image alignment and image similarity metrics.

| Metric (%) | LoRA | LoRA-Dash |
|---|---|---|
| Mean CLIP Text-Image Score | 32.0 | 32.8 |
| Mean Image Similarity | 76.4 | 78.2 |

While we try to ensure objectivity and fairness, it is worth noting that, unlike numerical comparisons, visual comparisons and user studies are inherently subjective. As for the actual effectiveness of LoRA-Dash, we look forward to feedback from researchers in a broader range of practical applications.

## E.2    MORE RESULTS FOR THE ALIGNMENT ON THE DIRECTIONS OF LORA-DASH

More results for Sec. 6.1 are shown in Tables. 21-22.

## E.3    MORE RESULTS FOR ABLATION STUDY

We here provide more results when fine-tuning DeBERTaV3-large on the ablation study in Fig. 10(b)-(c) in Sec. 6.4. The experimental results observed align well with the trends analyzed in Sec. 6.4, indicating consistency between our theoretical insights and practical outcomes.

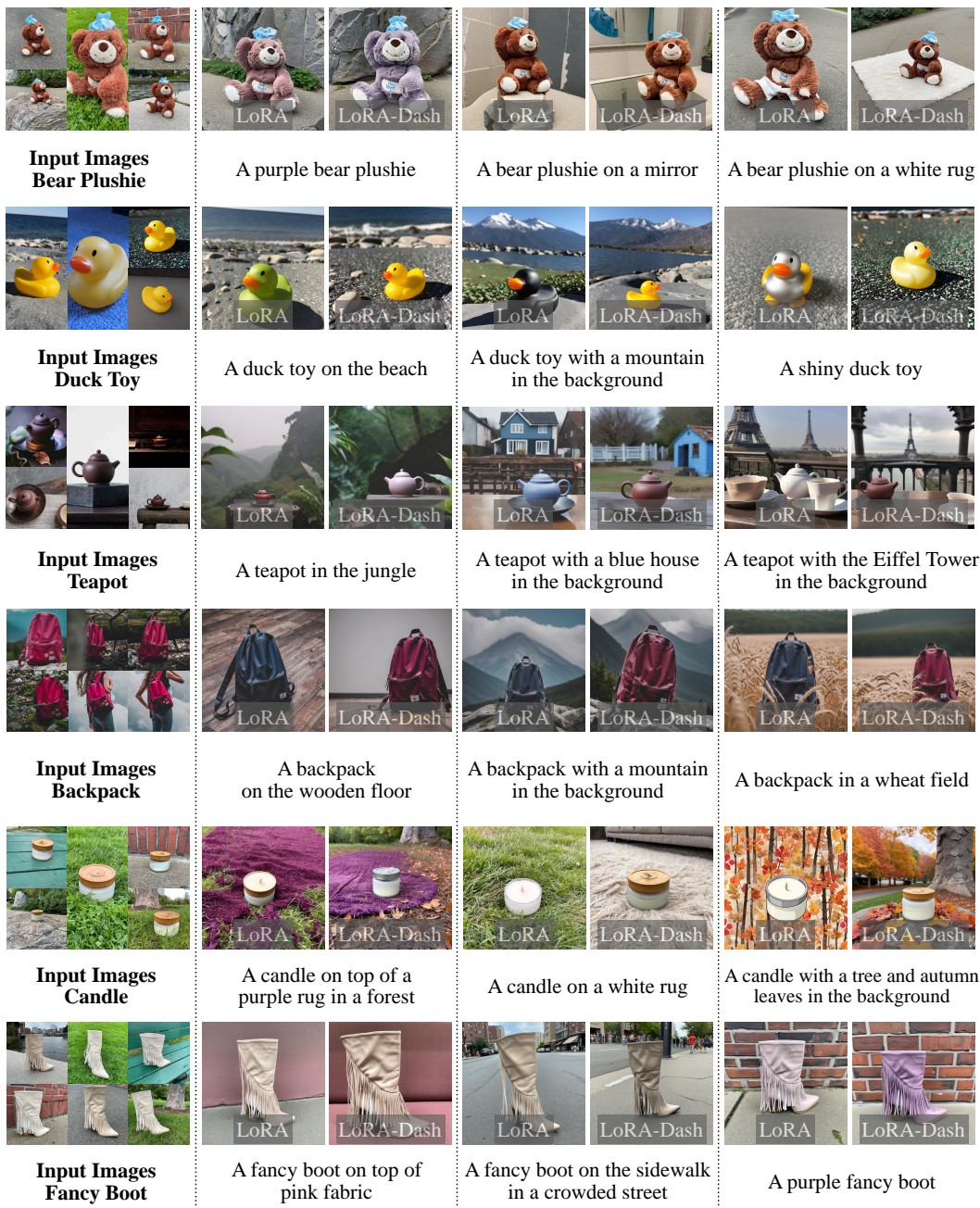

Figure 9: More results for subject-driven generation task.

We here further discuss the impact of selecting different dash directions. The effectiveness of choosing top, bottom, or random directions has been a topic frequently mentioned in various studies. However, we find differing conclusions in different works, which complicates the understanding. For instance, Meng et al. (2024) and Zhang & Pilanci (2024) suggest that selecting top directions yields the best results, while Hameed et al. (2024) advocates for random selections, and reference Wang et al. (2024) supports choosing the bottom. This inconsistency is puzzling and is a matter we aim to explore further through our research.

Indeed, fine-tuning leverages a model's broad pre-trained knowledge base, adapting it to learn specific, nuanced details relevant to particular downstream tasks. The essence of TSDs lies in their

Table 21: Alignment on the directions of LoRA-Dash with TSDs. We report the percentage from the self-attention value (v) projection of the first, middle and last layer of two models, as well as the average percentage of all q, k, and v modules.

| Model | LLaMA-7B ($r = 8$) | | | | DeBERTaV3-large ($r = 2$) | | | |
|---|---|---|---|---|---|---|---|---|
| Layer | First (1st) | Middle (16th) | Last (32nd) | All. | First (1st) | Middle (12nd) | Last (24th) | All. |
| $\Delta \mathbf{W}_{all} \cap \text{LTSD}$ | 1.00 | 0.75 | 1.00 | 0.74 | 1.00 | 1.00 | 1.00 | 0.99 |
| $\text{TSD} \cap \text{LTSD}$ | 1.00 | 1.00 | 1.00 | 0.78 | 0.75 | 0.75 | 0.50 | 0.73 |
| $\text{TSD} \cap \Delta \mathbf{W}_{all}$ | 1.00 | 0.75 | 1.00 | 0.83 | 0.75 | 0.75 | 0.50 | 0.76 |

Table 22: Alignment on the directions of LoRA-Dash with TSDs. We report the percentage from the self-attention value (v) projection of the first, middle and last layer of two models, as well as the average percentage of all q, k, and v modules.

| Model | LLaMA-7B ($r = 32$) | | | | DeBERTaV3-large ($r = 32$) | | | |
|---|---|---|---|---|---|---|---|---|
| Layer | First (1st) | Middle (16th) | Last (32nd) | All. | First (1st) | Middle (12nd) | Last (24th) | All. |
| $\Delta \mathbf{W}_{all} \cap \text{LTSD}$ | 0.75 | 0.75 | 0.75 | 0.75 | 0.75 | 1.00 | 1.00 | 0.97 |
| $\text{TSD} \cap \text{LTSD}$ | 0.75 | 0.75 | 0.75 | 0.77 | 0.75 | 0.75 | 1.00 | 0.78 |
| $\text{TSD} \cap \Delta \mathbf{W}_{all}$ | 1.00 | 1.00 | 1.00 | 0.81 | 0.75 | 1.00 | 1.00 | 0.82 |

representation of the singular directions where the original model weights exhibit the most significant changes during adaptation to new tasks. Intuitively, these TSDs, being the most altered, should indeed offer the most impactful directions for adjustment during fine-tuning, potentially yielding the best performance.

Analyzing the properties of TSDs reveals that they typically reside along the directions corresponding to smaller, yet not the smallest, singular values. If we choose to dash in the directions associated with the smallest singular values—the "bottom" directions—it's plausible that some of these might overlap with the TSDs. Hence, dashing in the bottom directions might produce results inferior to directly using the TSDs.

As for the top singular directions, which encapsulate the most significant information from pretraining, these generally involve more generalized knowledge. Since such information is less task-specific, adjusting these directions could potentially degrade performance rather than enhance it, especially if the task at hand requires nuanced understanding that diverges from the general pretraining data.

Lastly, a random selection of directions introduces unpredictability in outcomes. However, there is a chance that such random choices might inadvertently select appropriate directions or even TSDs, thus performing better than top selections but generally worse than bottom choices due to the randomness. This method could strike a balance between leveraging underlying model strengths and adapting to new specifics, albeit inconsistently.

In our empirical tests, the results largely align with our predictions. However, there are some exceptions where the effectiveness of top directions surpasses random or bottom, or where the bottom directions approach or even slightly exceed the performance of the TSDs. Please look for a counterexample in the Fig. 10(a). We consider these variations to be within the expected range, acknowledging that it's unrealistic to expect uniform trends across all tasks and models. The trends we have identified represent a more general, broadly applicable pattern.

In addition to the standard experimental setup, we conducted two additional experiments: adjusting the coordinate value changes of all directions or both top and bottom directions collectively. Neither of these configurations resulted in improved outcomes. This aligns with our analysis in Sec. 6.4, which suggested that the inclusion of irrelevant directions could degrade training performance due to increased noise in the model updates.

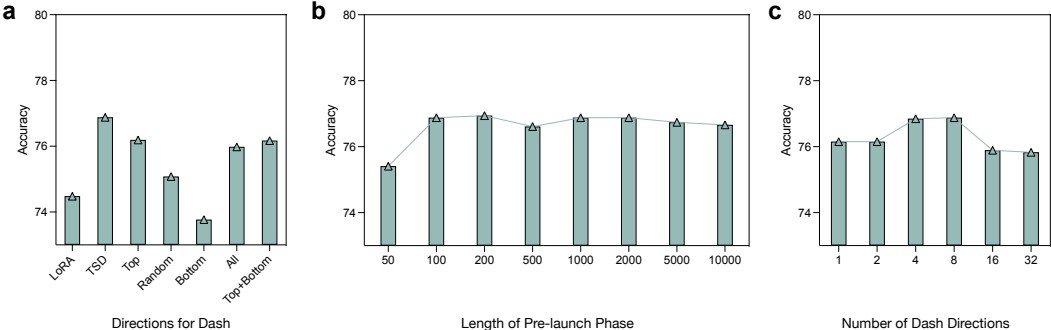

Figure 10: More results on CoLA when fine-tuning DeBERTaV3-large for ablation study of LoRA-Dash. **a**. Ablation study on the directions for dash. Beyond the top, random, and bottom directions, we also explored adjusting all directions and a combined adjustment of both bottom and top directions. **b**. Ablation study on the length of pre-launch phase $t$. **c**. Ablation study on the number of directions $s$ for dash.

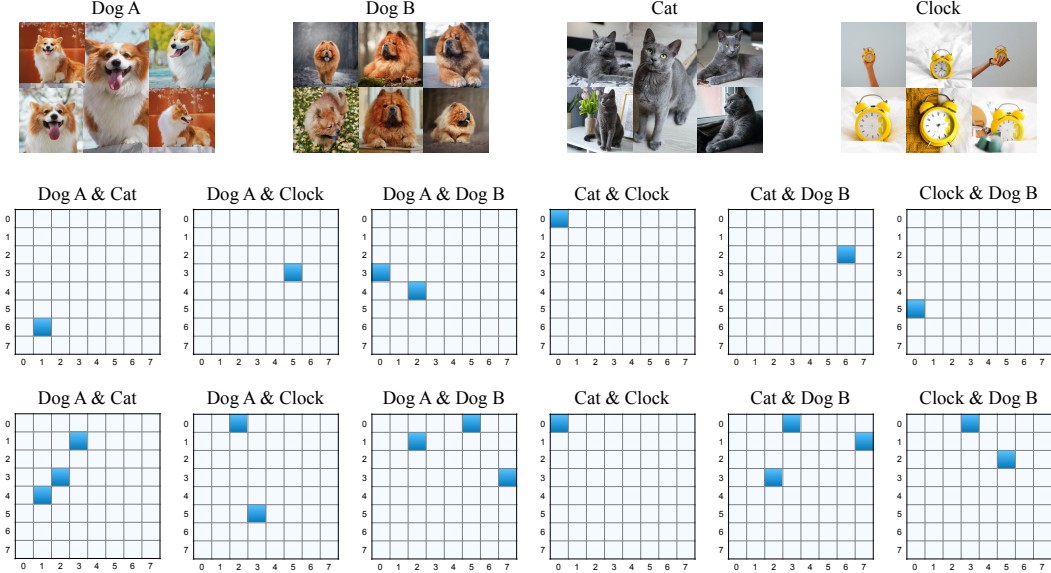

Figure 11: The results of the TSDs similarity between different tasks (subjects). For the first row: Images from different subjects. The second and third rows contain results extracted from two different layers of U-net Ronneberger et al. (2015). Each subplot illustrates the similarity of TSDs captured for two different subjects. The title of each subplot indicates the respective subjects, with the directional order of the first subject represented by columns and the second by rows. The deep blue blocks highlight where the directions for the two subjects coincide.

### E.4 ARE TSDS "TASK-SPECIFIC"?

We sought to verify whether TSDs are genuinely task-specific, that is, whether the most critical directions differ for each task. To investigate this, we employed a prototypical example: the subject-driven generation task. In this setting, we tracked the TSDs captured by LoRA-Dash when different subjects served as targets and analyzed the relationships between these TSDs. The results are presented in the Fig. 11 (Please refer to the caption of Fig. 11 for detailed information).

The experimental results affirm the task-specific nature of TSDs; indeed, each task exhibits distinct TSDs with minimal overlap between tasks. Moreover, even when tasks share the same direction,

its importance varies significantly between them. For instance, as depicted in the central subfigure of the first column in Fig. 11, both "Dog A" and "Cat" share a common direction, yet it ranks as the seventh most significant for "Dog A" and the second most significant for "Cat". Furthermore, tasks that are closely related tend to share more directions, as observed between "Dog A" and "Dog B"; however, the importance of these shared directions still differs for each task. This variability underscores the fundamentally task-specific property of TSDs, illustrating their unique and variable impact across different contexts.

### E.5 CONVERGENCE CURVE

We present loss curves for both LoRA-Dash and LoRA when fine-tuning LLaMA-7B. We ensure consistency in all training hyper-parameters, and the results are shown in Fig. 12(a). We can observe that during the pre-launch phase, the loss trajectories of LoRA-Dash and LoRA are aligned. However, as the training progresses into the dash phase, LoRA-Dash consistently exhibits a slightly lower loss compared to standard LoRA.

One may concern that a lower training loss could lead to overfitting. Therefore, we also tracked the validation loss throughout the training process, with the results presented in Fig. 12(b). It is clear that the validation loss for LoRA-Dash is much lower than that for LoRA. This suggests that LoRA-Dash not only avoids overfitting but also enhances its learning efficacy by effectively incorporating task-specific knowledge.

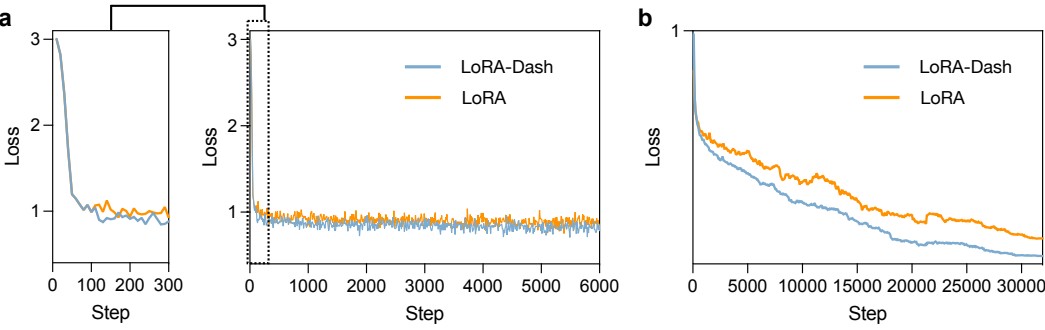

Figure 12: **a**. Training loss when fine-tuning LLaMA-7B. We record the training loss every 10 steps. **b**. Validation loss when fine-tuning LLaMA-7B. We record the training loss every 80 steps.

### E.6 COMPARISON WITH OTHER METHODS

In the main text, we only mainly compared the results of LoRA-Dash and LoRA because LoRA-Dash is designed to further harness the potential of TSDs captured by LoRA. Here, we further compare the performance of LoRA-Dash with other methods.

Specifically, we compare BitFit Zaken et al. (2021), Prompt Learning (Prefix) Li & Liang (2021), Series Adapter (Series) Houlsby et al. (2019), Adapter proposed by Pfeiffer et al. (2020) (PAdapter), Parallel Adapter (Parallel) He et al. (2021a), (IA)[3] Liu et al. (2022), SSL Si et al. (2024b), SSB Si et al. (2024b), AdaLoRA Zhang et al. (2022), PISSA Meng et al. (2024), MiLoRA Wang et al. (2024), FLoRA Si et al. (2024a) and DoRA Liu et al. (2024). We follow the settings detailed in our paper, and the results are shown in Tables. 23-24.

The results clearly show that even compared to state-of-the-art (SOTA) methods like DoRA and FLoRA, LoRA-Dash holds its ground impressively. Additionally, two phenomena are noteworthy:

- When fine-tuning LLaMA-7B, at a rank of 4, all methods except LoRA-Dash perform poorly under low parameter budgets, while LoRA-Dash achieves results comparable to those with higher parameter settings. This further highlights the superiority of LoRA-Dash, which stimulates performance on downstream tasks by unleashing the potential of TSDs.

Table 23: Results on commonsense reasoning tasks of LoRA-Dash compared with other methods. For LoRA derivatives, we report the parameter gains over LoRA.

| Method | Params | BoolQ | PIQA | SIQA | HellaS. | WinoG. | ARC-e | ARC-c | OBQA | Avg. |
|---|---|---|---|---|---|---|---|---|---|---|
| ChatGPT | - | 73.1 | 85.4 | 68.5 | 78.5 | 66.1 | 89.8 | 79.9 | 74.8 | 77.0 |
| *Fine-tuning LLaMA-7B* | | | | | | | | | | |
| Fully FT | 100% | 69.9 | 84.2 | 78.9 | 92.3 | 83.3 | 86.6 | 72.8 | 83.4 | 81.4 |
| Prefix | 0.11% | 64.3 | 76.8 | 73.9 | 42.1 | 72.1 | 72.9 | 54.0 | 60.6 | 64.6 |
| Series | 0.99% | 63.0 | 79.2 | 76.3 | 67.9 | 75.7 | 74.5 | 57.1 | 72.4 | 70.8 |
| Parallel | 3.54% | 67.9 | 76.4 | 78.8 | 69.8 | 78.9 | 73.7 | 57.3 | 75.2 | 72.2 |
| $\text{LoRA}_{r=4}$ | 0.10% | 2.3 | 46.1 | 18.3 | 19.7 | 55.2 | 65.4 | 51.9 | 57.0 | 39.5 |
| $\text{AdaLoRA}_{r=4}$ | + 0.6k | 66.1 | 78.1 | 74.3 | 34.0 | 74.4 | 76.7 | 57.5 | 71.2 | 66.5 |
| $\text{FLoRA}_{r=4}$ | + 2.6k | 67.2 | 78.0 | 72.9 | 65.4 | 73.8 | 73.8 | 55.3 | 71.8 | 69.8 |
| $\text{DoRA}_{r=4}$ | + 877k | 51.3 | 42.2 | 77.8 | 25.4 | 78.8 | 78.7 | 62.5 | 78.6 | 61.9 |
| $\text{LoRA-Dash}_{r=4}$ | + 1.3k | 65.2 | 79.9 | 78.3 | 82.8 | 77.1 | 78.6 | 65.4 | 78.4 | **75.7** |
| $\text{LoRA}_{r=32}$ | 0.83% | 68.9 | 80.7 | 77.4 | 78.1 | 78.8 | 77.8 | 61.3 | 74.8 | 74.7 |
| $\text{AdaLoRA}_{r=32}$ | + 5.1k | 69.1 | 82.2 | 77.2 | 78.3 | 78.2 | 79.7 | 61.9 | 77.2 | 75.5 |
| $\text{FLoRA}_{r=32}$ | + 164k | 66.4 | 81.3 | 77.1 | 75.6 | 77.1 | 77.2 | 62.4 | 77.6 | 74.3 |
| $\text{DoRA}_{r=32}$ | + 877k | 69.7 | 83.4 | 78.6 | 87.2 | 81.0 | 81.9 | 66.2 | 79.2 | 78.4 |
| $\text{LoRA-Dash}_{r=32}$ | + 1.3k | 69.9 | 82.8 | 78.6 | 84.9 | 81.6 | 82.3 | 66.5 | 80.8 | **78.4** |
| *Fine-tuning LLaMA3-8B* | | | | | | | | | | |
| $\text{LoRA}_{r=16}$ | 0.35% | 72.3 | 86.7 | 79.3 | 93.5 | 84.8 | 87.7 | 75.7 | 82.8 | 82.8 |
| $\text{AdaLoRA}_{r=16}$ | + 2.6k | 90.4 | 85.0 | 76.7 | 79.1 | 83.3 | 86.4 | 75.1 | 75.4 | 81.4 |
| $\text{FLoRA}_{r=16}$ | + 41k | 90.2 | 84.2 | 79.9 | 79.3 | 85.1 | 86.7 | 74.8 | 93.9 | 84.2 |
| $\text{LoRA-Dash}_{r=16}$ | + 1.3k | 74.8 | 88.0 | 80.6 | 95.2 | 85.6 | 89.0 | 77.4 | 84.8 | **84.4** |
| $\text{LoRA}_{r=32}$ | 0.70% | 70.8 | 85.2 | 79.9 | 91.7 | 84.3 | 84.2 | 71.2 | 79.0 | 80.8 |
| $\text{PISSA}_{r=32}$ | + 0 | 67.1 | 81.1 | 77.2 | 83.6 | 78.9 | 77.7 | 63.2 | 74.6 | 75.4 |
| $\text{MiLoRA}_{r=32}$ | + 0 | 68.8 | 86.7 | 77.2 | 92.9 | 85.6 | 86.8 | 75.5 | 81.8 | 81.9 |
| $\text{DoRA}_{r=32}$ | + 784k | 74.6 | 89.3 | 79.9 | 95.5 | 85.6 | 90.5 | 80.4 | 85.8 | 85.2 |
| $\text{LoRA-Dash}_{r=32}$ | + 1.3k | 75.3 | 88.5 | 80.2 | 95.7 | 86.8 | 90.7 | 80.2 | 85.6 | **85.4** |

Table 24: Results with DeBERTaV3-base fine-tuned on GLUE development set. "FT" represents fully fine-tuning.

| Method | % Params | MNLI Acc | SST-2 Acc | CoLA Mcc | QQP Acc | QNLI Acc | RTE Acc | MRPC Acc | STS-B Corr | All Avg. |
|---|---|---|---|---|---|---|---|---|---|---|
| FT | 100% | 89.90 | 95.63 | 69.19 | 91.87 | 94.03 | 83.75 | 90.20 | 91.60 | 88.27 |
| $(IA)^3$ | 0.03% | 89.44 | 95.52 | 67.01 | 89.01 | 91.80 | 79.42 | 88.23 | 90.79 | 86.40 |
| SSL | 0.02% | 88.35 | 95.07 | 66.64 | 88.19 | 90.10 | 82.31 | 88.68 | 90.13 | 86.18 |
| SSB | 0.05% | 89.86 | 95.53 | 67.82 | 89.87 | 93.41 | 83.75 | 88.72 | 90.94 | 87.49 |
| BitFit | 0.05% | 89.37 | 94.84 | 66.96 | 88.41 | 92.24 | 78.80 | 87.75 | 91.35 | 86.21 |
| Series | 0.17% | 90.10 | 95.41 | 67.65 | 91.19 | 93.52 | 83.39 | 89.25 | 91.31 | 87.73 |
| PAdapter | 0.16% | 89.89 | 94.72 | 69.06 | 91.05 | 93.87 | 84.48 | 89.71 | 91.38 | 88.02 |
| LoRA | 0.18% | 90.03 | 93.92 | 69.15 | 90.61 | 93.37 | 87.01 | 90.19 | 90.75 | 88.13 |
| AdaLoRA | 0.18% | 90.66 | 95.80 | 70.04 | 91.78 | 94.49 | 87.36 | 90.44 | 91.63 | 88.86 |
| FLoRA | 0.18% | 90.60 | 96.00 | 70.20 | 91.40 | 94.46 | 88.09 | 90.93 | 91.96 | 89.21 |
| DoRA | 0.22% | 90.21 | 94.38 | 69.33 | 90.84 | 93.26 | 86.94 | 90.19 | 91.34 | 88.31 |
| LoRA-Dash | 0.18% | 90.14 | 95.42 | 72.41 | 91.65 | 94.36 | 89.89 | 91.67 | 91.64 | **89.65** |

- Compared with LoRA derivatives, LoRA-Dash introduces significantly fewer additional parameters, such as nearly 130 times fewer than FLoRA and 700 times fewer than DoRA at rank 8 when fine-tuning LLaMA-7B, yet achieves better or comparable results.

Revisiting the objective of Parameter-Efficient Fine-Tuning, the key question is whether it is possible to achieve performance comparable to fully fine-tuning while training with significantly fewer parameters. Through extensive experimentation and comparison with other methods, LoRA-Dash maintains excellent performance even with a minimal parameter budget, where most methods would

be constrained by the amount of parameters available. This demonstrates that LoRA-Dash significantly surpasses other methods in terms of parameter utilization efficiency. Therefore, we conclude that LoRA-Dash exemplifies an ideal realization of PEFT goals.

*Reflecting on the entirety of this research, we arrive at a final conclusion:*

***"LoRA-Dash epitomizes the essence of parameter efficiency, maximizing the efficacy of each parameter deployed. With significantly fewer parameters trained, LoRA-Dash sustains or even exceeds the performance set by fully fine-tuning."***

*This marks the end of our current exploration, but it is merely a pause in the ongoing quest for deeper understanding. The path ahead remains rich with potential, inviting further inquiry into the intricate dance of knowledge...*

*The End*

