# OpenReview forum: "Unleashing the Power of Task-Specific Directions in Parameter Efficient Fine-tuning"
_ICLR.cc/2025/Conference — ICLR 2025 Poster_

### Official Review · Reviewer_27u4 · 2024-11-02

**Soundness:** 3
**Presentation:** 4
**Contribution:** 3
**Rating:** 6
**Confidence:** 4

**Summary:**

To enhance the prevalent parameter efficient fine-tuning (PEFT) strategies (e.g., LoRA) for large language models (LLMs), this paper delves into the concept of task-specific directions (TSDs) which are crucial for the adaptation of pre-trained LLMs to downstream tasks. Based on the proposed mathematical description of TSDs, a simple and effective scheme is designed to approximately identify TSDs from the parameter difference matrix obtained through LoRA. The model parameters derived from LoRA are then aligned with the estimated TSDs to enhance LoRA's performance. The results of extensive experiments confirm the validity of the mathematical description and the effectiveness of the proposed PEFT framework across diverse learning settings.

**Strengths:**

1. This paper is well organized and easy to follow.
2. In contrast to the conceptual discussion of task-specific directions (TSDs) in existing parameter-efficient fine-tuning methods, this paper offers a deeper understanding by providing a mathematical description of TSDs, which is crucial for enhancing the performance of these parameter-efficient fine-tuning methods.
3. Te proposed TSDs identification and utilization method are simple and effective.
4. Experimental results across diverse settings validate the significance of task-specific directions in fine-tuning large language models (LLMs), and demonstrate the effectiveness of the proposed LoRA-Dash compared to the prevalent LoRA algorithm.

**Weaknesses:**

1. As a formal definition, the description of task-specific directions (i.e., “directions whose coordinate values exhibit significantly higher change rates $\delta$ through alteration” in Definition 4) is somewhat vague. The term “significantly higher change rate” would benefit from a more precise definition, such as being specified with reference to a particular threshold.
2. More discussions about how to select the value of hyper-parameter $s$ should be included, as the choice of $s$ directly determines the exact task-specific directions.
3. The statement in Proposition 1 is supported solely by observations from experiments and is not accompanied by rigorous proof.
4. The evaluation results of Fully FT method regarding LLaMA2-7B and LLaMA3-7B models are missed in Table 1.
5. To thoroughly evaluate and compare the effects of task-specific directions and other core directions, it is recommended to include the vanilla LoRA as a baseline method in Figure 5a.
6. The primary contribution of this paper is the proposal of a simple and effective improvement scheme for the LoRA algorithm and its variants. Although this contribution is commendable for acceptance, its impact on the broader research community is still limited, which hinders me from assigning a higher rating.

**Questions:**

Please refer to weaknesses 1-5 listed in the section above.

---

> ### Author Response · Authors · 2024-11-16
> **Response to Reviewer 27u4**
>
> Dear Reviewer 27u4:
>
> We would like to first extend our sincere gratitude for your time and effort in evaluating our manuscript. Your thorough evaluation and insightful comments are greatly appreciated. We will address your questions point by point and hope to resolve your concerns effectively.
>
>
>
> ### **Weakness 1, Vague Definition**
>
> Thank you for your valuable suggestion. Indeed, we have given this definition considerable thought. Determining the number of TSDs is analogous to selecting $k$ in the K-means algorithm. However, we can modify the manuscript to define TSDs as the top-s directions with the largest rate of change, as specifying a fixed number is more intuitive than setting a threshold.
>
>
>
> ### **Weakness 2，Hyper-parameter $s$ Setting**
>
> Thanks for your suggestions. In our experiment, we set $s=8$, which is consistent with those in the observations for the sake of overall uniformity. Regarding the selection of the hyper-parameter $s$, we found in our ablation experiments that the model’s performance remains robust across a wide range of $s$. This robustness is due to our approach of selecting $s$ directions based on the ranking of their change rates. As a result, regardless of the value of $s$, we always focus on the most significant directions. Therefore, for various scenarios, setting $s=8$ ensures consistently excellent performance.
>
> However, we observed a performance drop when $s$ becomes too large. Based on our analysis in Sec. D.7, we suspect that the inclusion of some irrelevant directions might introduce noise into the training process, destabilizing model convergence and ultimately leading to diminished performance.
>
>
>
> ### **Weakness 3, Rigorous Proof**
>
> We fully agree with your perspective. In practice, for fine-tuning, it is not feasible to obtain the optimal $\mathbf{W}^*$, let alone analyze its properties or provide rigorous mathematical proofs. Therefore, we can only rely on the statistical patterns observed through experiments to validate our propositions.
>
>
>
> ### **Weakness 4, Results of Fully FT**
>
> Thanks for your suggestions. We have added the results of Fully FT methods regarding LLaMA2-7B and LLaMA3-8B in the updated manuscript.
>
>
>
> ### **Weakness 5, Figure Refinement**
>
> Thanks for your suggestions. We have updated Fig. 5(a) in the manuscript.
>
>
>
> ### **Weakness 6, Contribution**
>
> We appreciate your perspective on the contributions of our work. However, we would like to emphasize that the primary focus of this paper lies in the exploration and study of TSD, as highlighted in Appendix A, as well as a novel PEFT method.
>
> Indeed, TSD is not only critical for PEFT but also plays a significant role in other areas such as Multi-Task Learning (MTL). In modern large-scale model training, multi-task learning has become inevitable. Multi-task learning often suffers from task interference, where learning one task degrades performance on others.  However, exploring TSD information allows us to identify the most important directions for each specific task, helping mitigate this issue by isolating task-relevant adaptations, thus maintaining task boundaries. By focusing on task-specific directions, we can also allocate computational and model resources more effectively.
>
> Therefore, TSD provides valuable insights for the broader research community. It serves as one of the key contributions of our work.

---

> > ### Comment · Reviewer_27u4 · 2024-12-03
> >
> > Thank you for the authors' detailed responses. most of my concerns have been addressed. Therefore, I would like to maintain my positive rating.

---

### Official Review · Reviewer_KtSW · 2024-11-03

**Soundness:** 3
**Presentation:** 3
**Contribution:** 3
**Rating:** 6
**Confidence:** 4

**Summary:**

This paper addresses the resource-intensive issue of fully fine-tuning large language models. It focuses on task-specific directions (TSDs) in parameter-efficient fine-tuning (PEFT). LoRA is examined, and its lack of a clear TSD definition is noted. A framework is then introduced to define TSDs based on the relationship between pre-trained weights and optimal weights for specific tasks. The properties of TSDs are explored, and challenges in using them are identified. Despite the unknowns in practical fine-tuning, it's found that LoRA's ∆W can capture TSD information. The novel LoRA-Dash method is proposed, comprising pre-launch and dash phases. Experiments show that LoRA-Dash outperforms LoRA, is robust to parameter budgets, and excels in various tasks. It also enhances other PEFT methods and provides valuable insights for optimizing model fine-tuning.

**Strengths:**

- A task-aware approach for identifying critical gradient directions is proposed.

- An effective task-specific fine-tuning scheme Lora-Dash has been introduced.

- Theoretical analysis and extensive experiments have been conducted, providing comprehensive validation of the effectiveness of the proposed algorithm.

**Weaknesses:**

- More justification is expected. For example, does fine-tuning a general-purpose model for a specific task risk compromising its generalization capability? It is essential to explore strategies that enable a model, even after fine-tuning, to retain the flexibility to handle a range of general tasks. Striking a balance between task-specific adaptation and broad applicability remains a key challenge in fine-tuning.

- The experiments can use more perspectives. For example, the observations in Figure 1 are based solely on commonsense reasoning tasks in the LLama model. Further comparisons with other relevant methods (such as Lora-ga，dora) and other models (such as Qwen, Mistral) should be included for a more comprehensive analysis.

- In the visualization part, as shown in Figure 4 and Figure 8, a comparison with the results from full fine-tuning should also be provided.

**Questions:**

Mostly the above comments. Also some minors:

- Is Lora-Dash applicable to multimodal models, such as LLaVA?

- Why does the performance of LoRA-Dash decline when the rank increases beyond a certain point?

- How can we address the issue of excessive memory requirements for SVD in the application of this method?

---

> ### Author Response · Authors · 2024-11-13
> **Response to Reviewer KtSW**
>
> Dear Reviewer KtSW:
>
> We would like to first extend our sincere gratitude for your time and effort in evaluating our manuscript. Your thorough evaluation and insightful comments are greatly appreciated. We will address your questions point by point and hope to resolve your concerns effectively.
>
>
> ### **Weakness 1, Generalization Capability**
>
> We appreciate your perspective that fine-tuned models should also retain generalization capabilities. However, we would like to point out that in the context of PEFT, the primary objective is to efficiently adapt a model to a specific downstream task. Under this setup, the focus is solely on achieving strong performance on the downstream task, without prioritizing whether the model retains its generalization capability beyond that task.
>
> As for the idea of “exploring strategies that enable a model, even after fine-tuning, to retain the flexibility to handle a range of general tasks,” we believe this is not closely aligned with the core focus of PEFT research. Instead, it is more related to incremental learning, which aims to enable a model to learn new tasks or knowledge without forgetting previously acquired knowledge. These are two distinct research areas with different objectives and methodologies.
>
>
>
> ### **Weakness 2, More Perspective**
>
> We  agree with your suggestion that using more models would indeed enhance the generality of our observations. We will conduct corresponding experiments on additional models and provide you with feedback as soon as possible.
>
>
>
> ### **Weakness 3, Figure Refine**
>
> Thank you for your suggestion. We will add results from full fine-tuning in Figures 4 and 8.
>
>
>
> ### **Question 1, Applicable to Multimodal Models.**
>
> LoRA-Dash is indeed capable of fine-tuning multimodal models. In fact, it can leverage the information from TSD to better achieve multimodal alignment. We are currently working on related tasks and progressing in this direction.
>
>
>
> ### **Question 2, Performance Decline**
>
> It is not surprising that PEFT methods, such as LoRA-Dash, exhibit performance degradation as the parameter budget (which can be evaluated by rank) increases. In fact, this phenomenon is quite common. While an increase in the number of parameters enhances the expressiveness of the method, it also introduces training instability, which can potentially lead to performance drops. Additionally, [1] theoretically demonstrates that the larger the parameter count in PEFT fine-tuning, the more the model’s stability and generalization ability tend to decline.
>
> [1] 2023, AAAI, On the Effectiveness of Parameter-Efficient Fine-Tuning
>
>
>
> ### **Question 3, Memory Requirements for SVD**
>
> This is an excellent and practical question. We propose two strategies to address this issue:
>
> 1. Two-Stage Strategy: Perform SVD decomposition on the pre-trained weights before the training starts and save the results. During the training process, these precomputed results can be directly utilized, eliminating the need for on-the-fly decomposition.
>
> 2. On-the-Fly Decomposition: Perform the SVD decomposition on the CPU during the training process. This approach offloads the computational burden from the GPU, ensuring the primary training workflow remains efficient.
>
> Both strategies provide practical solutions depending on resource availability and training requirements.
>
> Additionally, many existing works also utilize SVD decomposition on pre-trained weights [2-4], and the aforementioned strategies are equally applicable to these methods.
>
> [2] 2024, NIPS, Spectral Adapter: Fine-Tuning in Spectral Space
>
> [3] 2024, NIPS, PiSSA: Principal Singular Values and Singular Vectors Adaptation of Large Language Models
>
> [4] 2023, ICCV, SVDiff: Compact Parameter Space for Diffusion Fine-Tuning
>
> We sincerely hope that we can address your concerns.

---

> ### Author Response · Authors · 2024-11-16
> **Response to Reviewer KtSW**
>
> Dear Reviewer KtSW,
>
> we have already updated the results related to Weaknesses 2 and 3.
>
> Weakness 2: We fine-tune Qwen2.5-7B on math reasoning task, and the results are shown in Fig. 7 in Appendix.
>
> Weakness 3: We compare the images generated by LoRA-Dash and fully fine-tuning, and the results are shown in supplementary. Since each dataset in this task contains only a few images, fully training a stable diffusion model is highly prone to overfitting. As a result, most works primarily compare with the results of parameter efficient fine-tuning [1,2]. However, as requested by the Reviewer KtSW, we also present the results of fully training the model. Unsurprisingly, the performance of the fully trained model is worse than LoRA-Dash, as expected.
>
> [1] 2024, arXiv, NoRA: Nested Low-Rank Adaptation for Efficient Fine-Tuning Large Models
>
> [2] 2024, EMNLP, Mixture-of-subspaces in Low-rank Adaptation

---

> > ### Comment · Reviewer_KtSW · 2024-12-03
> >
> > Thank the authors for the response to the concerns and and questions. I am good with the positive rating.

---

### Official Review · Reviewer_B9L3 · 2024-11-04

**Soundness:** 3
**Presentation:** 3
**Contribution:** 3
**Rating:** 6
**Confidence:** 3

**Summary:**

This paper critiques LoRA's prior TSD exploration and provides a precise TSD definition to better understand their role in fine-tuning large language models. The authors introduce LoRA-Dash, which fully exploits TSDs potential, and demonstrate its significant advantages over conventional methods through extensive experimentation. The findings validate LoRA-Dash's effectiveness and highlight the importance of TSDs in parameter-efficient fine-tuning, aiming to inspire further advancements in natural language processing and beyond.

**Strengths:**

1. The authors observe that TSD can be predicted from delta W in LoRA-based fine-tuning, offering a new perspective on task-specific directions.
2. This paper provides a precise definition of task-specific directions and explores their application in LoRA-based parameter-efficient fine-tuning.
3. The authors propose the LoRA Dash algorithm, which proactively utilizes these influences to enhance the model fine-tuning process.

**Weaknesses:**

See detailed questions.

**Questions:**

Thanks for submitting to ICLR'25, I really enjoy reading your paper. I think your paper focuses on a hot and important topic. However, I have several further questions:

1. The paper lacks evaluation of the algorithm's overhead on end-to-end fine-tuning time. Can LoRA-Dash accelerate the fine-tuning phase as suggested by its name?
2. The benefits of the LoRA-Dash algorithm are unclear. Although it outperforms the vanilla LoRA approach when r is small, its improvement over the best fine-tuned LoRA model is minimal. Please clarify the real benefits of the LoRA-Dash algorithm.
3. Can you provide insights on applying TSD observations to other parameter-efficient fine-tuning methods, such as adapter/P-tuning? Are these methods suitable for this approach?

---

> ### Author Response · Authors · 2024-11-16
> **Response to Reviewer B9L3**
>
> Dear Reviewer B9L3:
>
> We would like to first extend our sincere gratitude for your time and effort in evaluating our manuscript. Your thorough evaluation and insightful comments are greatly appreciated. We will address your questions point by point and hope to resolve your concerns effectively.
>
>
>
> ### **Question 1, LoRA-Dash Fine-tuning Time**
>
> Thank you for your valuable suggestion. We apologize if the term “Dash” may have caused any misunderstanding. Here, “Dash” does not refer to training time acceleration; rather, it represents the capability to accelerate convergence. By expediting the learning of TSD information, LoRA-Dash achieves significantly faster convergence compared to LoRA. In practical fine-tuning scenarios, such as with DeBERTaV3-large, LoRA-Dash requires only half the number of training epochs needed by LoRA, demonstrating its efficiency and effectiveness.
>
>
>
> ### **Question 2, Benefits of LoRA-Dash**
>
> In PEFT, the primary objective, from the perspective of evaluation metrics, is to achieve better performance with fewer parameters. As you suggested, LoRA-Dash demonstrates its superior performance by achieving better results with a very small rank $r$, which is a clear reflection of its outstanding effectiveness.
>
> As $r$ increases, the number of fine-tuned parameters grows, and the performance of fine-tuning methods naturally approaches that of fully fine-tuning. In this scenario, LoRA-Dash cannot exhibit the same significant performance gains over LoRA as observed with smaller parameter budgets, as the upper limit of performance is constrained. However, it still achieves noticeable improvements over LoRA. This further highlights the superiority of LoRA-Dash in effectively utilizing the parameter budget to enhance performance.
>
>
>
> ### **Question 3, Applying TSD to Other Methods**
>
> This is an excellent and thought-provoking question. Below is our perspective:
>
> Within the framework we have established, TSD represents the direction in which model weights require substantial modifications. Therefore, the application of TSD essentially involves modifying the corresponding model weights. From this perspective, methods such as P-Tuning, which focus on modifying the input, are less suitable for applying TSD. On the other hand, approaches like Adapter and LoRA are naturally more compatible with TSD.
>
> Furthermore, when practically applying TSD, we need to leverage the information of $\Delta \mathbf{W}$, which represents the weight changes. For LoRA and its variants, TSD is inherently compatible, as these methods can be regarded as learning  $\Delta \mathbf{W}$ directly. In contrast, for methods like Adapter, it is necessary to record the pre-trained weights and compute the difference between the updated weights during fine-tuning and the pre-trained weights to obtain $\Delta \mathbf{W}$. This additional step is required to effectively integrate TSD into these approaches.
>
> We sincerely hope that our response could address your concerns.

---

> > ### Comment · Reviewer_B9L3 · 2024-11-24
> >
> > Thank you for the author's rebuttal. I have no further comments to add to your response. I appreciate the clarification, particularly regarding the benefits of LoRA-Dash compared to LoRA. Thank you.

---

### Official Review · Reviewer_93T4 · 2024-11-05

**Soundness:** 3
**Presentation:** 3
**Contribution:** 2
**Rating:** 5
**Confidence:** 4

**Summary:**

This paper proposes a parameter-efficient fine-tuning (PEFT) approach, LoRA-Dash, that explicitly exploits task-specific directions to improve parameter-efficiency. The proposed method consists of two phases: in the pre-launch phase, LoRA fine-tuning is performed for a certain number of steps to identify task-specific directions, which are estimated as the core directions of the pre-trained matrix most amplified by the LoRA update matrix. In the dash phase, changes in these directions are explicitly parameterized alongside the LoRA matrices to further enhance task alignment. The authors empirically demonstrate that LoRA-Dash outperforms standard LoRA and recent PEFT methods across a range of tasks and base models and show the robustness of the estimated task-specific directions.

**Strengths:**

A substantive assessment of the strengths of the paper, touching on each of the following dimensions: originality, quality, clarity, and significance. We encourage reviewers to be broad in their definitions of originality and significance. For example, originality may arise from a new definition or problem formulation, creative combinations of existing ideas, application to a new domain, or removing limitations from prior results. You can incorporate Markdown and Latex into your review. See https://openreview.net/faq.
1. This paper introduces a novel framework for identifying task-specific directions in pre-trained models by projecting various task-related matrices onto the subspace of the original weight matrix, providing a consistent basis for analyzing task-specific adaptations.
2. The proposed method effectively leverages task-specific directions, achieving significant improvements over LoRA with minimal computational overhead, demonstrating its practical advantage as a PEFT method.
3. Empirical evaluations show that the task-specific directions derived from this approach align with those amplified by full fine-tuning, indicating that the method successfully identifies important directions for task adaptation.

**Weaknesses:**

A substantive assessment of the weaknesses of the paper. Focus on constructive and actionable insights on how the work could improve towards its stated goals. Be specific, avoid generic remarks. For example, if you believe the contribution lacks novelty, provide references and an explanation as evidence; if you believe experiments are insufficient, explain why and exactly what is missing, etc.
1. [Reliance on Preliminary Experiments for Hyperparameter Selection]
The proposed method relies on preliminary experiments to set key hyperparameters, such as the length of pre-launch phase and the number of dash directions. Although the chosen hyperparameters work well on the evaluated benchmarks, there is no guarantee that these settings generalize to other tasks and models. Moreover, the authors do not provide a systematic or principled approach for selecting these hyperparameters in varying scenarios.
2. [Lack of Objective Evaluation for Diffusion Models]
The evaluation of diffusion models relies solely on subjective assessments, which introduces potential biases and limits reproducibility. Incorporating quantitative metrics for diffusion model performance would strengthen the evaluation and provide a more comprehensive view of the model’s effectiveness.

**Questions:**

1. In Figure 5, the performance on OBQA with TSD in the left-most subplot does not match with Length of Pre-launch Phase set to 100 in the middle subplot. Why is this the case since both settings should correspond to the setting used in the main experiment?
2. For subject-driven generation tasks, how do the task-specific directions correspond to the objects or features in the image?

---

> ### Author Response · Authors · 2024-11-16
> **Response to Reviewer 93T4**
>
> Dear Reviewer 93T4:
>
> We would like to first extend our sincere gratitude for your time and effort in evaluating our manuscript. Your thorough evaluation and insightful comments are greatly appreciated. We will address your questions point by point and hope to resolve your concerns effectively.
>
>
>
> ### **Weakness 1,  Hyper-parameter Selection**
>
> Our choice of hyper-parameters is not necessarily tied to the preliminary observations. We set the hyper-parameters to be consistent with those in the observations for the sake of overall uniformity. We also conducted hyper-parameter ablation experiments across multiple models, including LLaMA3-8B, LLaMA-7B, and DeBERTaV3-large, on different tasks. Beyond the results presented in the main text, additional findings are provided in the Appendix on page 30, Figure 10. These results show that when hyper-parameters fall within a reasonable range, the performance of the model remains relatively stable. Therefore, for various scenarios, setting $s=8$ ensures consistently excellent performance.
>
> Therefore, we believe that using default parameters can achieve satisfactory results across different scenarios, without the need to design specific hyper-parameters for each case. Moreover, in all experiments, our hyper-parameter settings were kept consistent, further demonstrating the robustness of LoRA-Dash, which does not require different hyper-parameter choices for varying scenarios.
>
>
>
> ### **Weakness 2, Objective Evaluation for Diffusion Models**
>
> Thank you for your valuable feedback. We fully acknowledge your perspective regarding the reliance on subjective assessments in generative experiments, as we also mentioned in Appendix E.1, Line 1445, “unlike numerical comparisons, visual comparisons and user studies are inherently subjective.” To ensure objectivity as much as possible, we conducted a user study, the results of which are presented in Appendix E.1. The study shows that the images generated by LoRA-Dash achieved an approval rating of 84.51%.
>
> Additionally, such user studies are a commonly adopted quantitative evaluation strategy for generative results in PEFT works [1-3]. Moreover, to fully address your concerns, we have also performed the CLIP  text-image score, which measures the alignment between textual descriptions and corresponding images, and image similarity score, which assesses the similarity between the original image and the generated image. The results are shown in the following table, where it is evident that for both metrics, LoRA-Dash outperforms LoRA. The images generated by LoRA-Dash exhibit better text alignment as well as stronger correspondence with the original images. This indicates that LoRA-Dash produces superior results compared to LoRA. We will add these results to our paper, thank you for your suggestions.
>
> | Metric （%）               | LoRA | LoRA-Dash |
> | -------------------------- | ---- | --------- |
> | Mean CLIP Text-Image Score | 32.0 | 32.8      |
> | Mean Image Similarity      | 76.4 | 78.2      |
>
> [1] 2024, EMNLP, Mixture-of-Subspaces in Low-Rank Adaptation
>
> [2] 2024, arxiv, LoRA-Composer: Leveraging Low-Rank Adaptation for Multi-Concept Customization in Training-Free Diffusion Models
>
> [3] 2024, ECCV, Concept Sliders: LoRA Adaptors for Precise Control in Diffusion Models
>
>
>
> ### **Question 1, Misalignment of the Experiment Results**
>
> Figure 5(a) presents the ablation experiment results on LLAMA3-8B, while (b) and (c) show results on LLaMA-7B. Therefore, the performance differences on OBQA are due to the variations in model size and architecture.
>
>
>
> ### **Question 2, Relations between TSDs and Objects**
>
> This is an excellent question. We have discussed it in Appendix E.4 on page 30 and hope it resolves your concerns.

---

### Author Response · Authors · 2024-11-16
**Common Response**

Dear AC and all reviewers,

We authors sincerely appreciate the time and effort you have dedicated to reviewing our work, and we are also deeply grateful for the valuable suggestions provided by the reviewers, which have significantly improved the quality of our manuscript. In response to the reviewers' constructive feedback, we have made the following revisions to our paper:

1. In response to Reviewer 93T4, Weakness 2, we have added an objective evaluation of visual comparisons. This update can be found in Appendix E.1, Lines 1430–1444.

2. In response to Reviewer KtSW, Weakness 2, we have included fine-tuning experiments on Qwen 2.5-7B for math reasoning task to further demonstrate the universality of TSD. These results are presented in Appendix Figure 7.

3. In response to Reviewer 27u4, Weakness 4, we have added experimental results of fully fine-tuning LLaMA2-7B and LLaMA3-8B on commonsense reasoning tasks. These results are included in Table 1 of the main text.

4. In response to Reviewer 27u4, Weakness 5, we have expanded the ablation study to include a comparison with LoRA. This update is reflected in Figure 5a of the main text.

We hope these revisions address the concerns raised by the reviewers and further clarify and strengthen our work. Thank you again for your insightful feedback and guidance.

Sincerely,

Authors.

---

### Author Response · Authors · 2024-11-22

Dear Reviewers,

May we kindly ask if our responses have addressed your concerns? We look forward to further discussions and feedback from you!

Sincerely,

Authors

---

### Meta-Review · Area_Chair_X7HA · 2024-12-20

**Metareview:**

The paper proposes the concept of task-specific directions in fine-tuning LLMs, modifying LoRA to push modifications wherever such directions point towards. Experiments validate that this improves finetuning performance on targeted tasks.

Among the paper's strengths is that this task-specific directions approach lead to performance comparable to full fine-tuning on the chosen tasks, which is quite impressive. Several concerns raised by reviewers were addressed (see below). Though the paper has improved overall, there are still some parts that require polishing.

**Additional Comments On Reviewer Discussion:**

Authors addressed several reviewer concerns during the rebuttal phase, including: (a) visual comparisons were coupled with an objective evaluation,  fine-tuning experiments where further applied to Qwen 2.5-7B over a  math reasoning task, and to  LLaMA2-7B and LLaMA3-8B on commonsense reasoning tasks, as well as an ablation study to include a comparison with LoRA. These have improved the paper overall.

A reviewer pointed out the statement in Proposition 1 is supported solely by observations from experiments. It is not common to call such a claim a "Proposition".  The authors should refer to this as an "observation", but they can simply discuss what they observe without giving it a "lemma" form. The language around this section should also be toned down/made precise and less colloquial; the proposition should not be referred to as "exhilarating", and statements like "we remain undeterred and placed our hopes on ∆W" should be rephrased.

---

### Decision · Program_Chairs · 2025-01-22

Accept (Poster)